# FEDFA: FEDERATED FEATURE AUGMENTATION

**Tianfei Zhou & Ender Konukoglu**
Computer Vision Lab, ETH Zurich

## ABSTRACT

Federated learning is a distributed paradigm that allows multiple parties to collaboratively train deep models without exchanging the raw data. However, the data distribution among clients is naturally non-i.i.d., which leads to severe degradation of the learnt model. The primary goal of this paper is to develop a robust federated learning algorithm to address *feature shift* in clients' samples, which can be caused by various factors, *e.g.*, acquisition differences in medical imaging. To reach this goal, we propose FEDFA to tackle federated learning from a distinct perspective of federated feature augmentation. FEDFA is based on a major insight that each client's data distribution can be characterized by statistics (*i.e.*, mean and standard deviation) of latent features; and it is likely to manipulate these local statistics *globally*, *i.e.*, based on information in the entire federation, to let clients have a better sense of the underlying distribution and therefore alleviate local data bias. Based on this insight, we propose to augment each local feature statistic probabilistically based on a normal distribution, whose mean is the original statistic and variance quantifies the augmentation scope. Key to our approach is the determination of a meaningful Gaussian variance, which is accomplished by taking into account not only biased data of each individual client, but also underlying feature statistics characterized by all participating clients. We offer both theoretical and empirical justifications to verify the effectiveness of FEDFA. Our code is available at https://github.com/tfzhou/FedFA.

## 1 INTRODUCTION

Federated learning (FL) (Konečnỳ et al., 2016) is an emerging collaborative training framework that enables training on decentralized data residing devices like mobile phones. It comes with the promise of training centralized models using local data points such that the privacy of participating devices is preserved, and has attracted significant attention in critical fields like healthcare or finance. Since data come from different users, it is inevitable that the data of each user have a different underlying distribution, incurring large heterogeneity (non-iid-ness) among users' data. In this work, we focus on *feature shift* (Li et al., 2020b), which is common in many real-world cases, like medical data acquired from different medical devices or natural image collected in diverse environments.

While the problem of feature shift has been studied in classical centralized learning tasks like domain generalization, little is understood how to tackle it in federated learning. (Li et al., 2020b; Reisizadeh et al., 2020; Jiang et al., 2022; Liu et al., 2020a) are rare exceptions. FEDROBUST (Reisizadeh et al., 2020) and FEDBN (Li et al., 2020b) solve the problem through *client-dependent* learning by either fitting the shift with a client-specific affine distribution or learning unique BN parameters for each client. However, these algorithms may still suffer significant local dataset bias. Other works (Qu et al., 2022; Jiang et al., 2022; Caldarola et al., 2022) learn robust models by adopting Sharpness Aware Minimization (SAM) (Foret et al., 2021) as the local optimizer, which, however, doubles the computational cost compared to SGD or Adam. In addition to model optimization, FEDHARMO (Jiang et al., 2022) has investigated specialized image normalization techniques to mitigate feature shift in medical domains. Despite the progress, there leaves an alternative space – *data augmentation* – largely unexplored in federated learning, even though it has been extensively studied in centralized setting to impose regularization and improve generalizibility (Zhou et al., 2021; Zhang et al., 2018).

While seemingly straightforward, it is non-trivial to perform effective data augmentation in federated learning because users have no direct access to external data of other users. Simply applying conventional augmentation techniques to each client is sub-optimal since without injecting global

information, augmented samples will most likely still suffer local dataset bias. To address this, FED-MIX (Yoon et al., 2021) generalizes MIXUP (Zhang et al., 2018) into federated learning, by mixing averaged data across clients. The method performs augmentation in the input level, which is naturally weak to create complicated and meaningful semantic transformations, *e.g.*, *make-bespectacled*. Moreover, allowing exchange of averaged data will suffer certain levels of privacy issues.

In this work, we introduce a novel federation-aware augmentation technique, called FedFA, into federated learning. FEDFA is based on the insight that statistics of latent features can capture essential domain-aware characteristics (Huang & Belongie, 2017; Zhou et al., 2021; Li et al., 2022a;b; 2021a), thus can be treated as "features of participating client". Accordingly, we argue that the problem of feature shift in FL, no matter the shift of each local data distribution from the underlying distribution, or local distribution differences among clients, even test-time distribution shift, can be interpreted as the shift of feature statistics. This motivates us to directly addressing local feature statistic shift by incorporating universal statistic characterized by all participants in the federation.

FEDFA instantiates the idea by online augmenting feature statistics of each sample during local model training, so as to make the model robust to certain changes of "features of participating client". Concretely, we model the augmentation procedure in a probabilistic manner via a multivariate Gaussian distribution. The Gaussian *mean* is fixed to the original statistic, and *variance* reflects the potential local distribution shift. In this manner, novel statistics can be effortlessly synthesized by drawing samples from the Gaussian distribution. For effective augmentation, we determine a reasonable variance based on not only variances of feature statistics within each client, but also universal variances characterized by all participating clients. The augmentation in FEDFA allows each local model to be trained over samples drawn from more diverse feature distributions, facilitating local distribution shift alleviation and client-invariant representation learning, eventually contributing to a better global model.

FEDFA is a conceptually simple but surprisingly effective method. It is non-parametric, requires negligible additional computation and communication costs, and can be seamlessly incorporated into arbitrary CNN architectures. We propose both theoretical and empirical insights. Theoretically, we show that FEDFA implicitly introduces regularization to local model learning by regularizing the gradients of latent representations, weighted by variances of feature statistics estimated from the entire federation. Empirically, we demonstrate that FEDFA (1) works favorably with extremely small local datasets; (2) shows remarkable generalization performance to *unseen* test clients outside of the federation; (3) outperforms traditional data augmentation techniques by solid margins, and can complement them quite well in the federated learning setup.

## 2 OUR APPROACH

### 2.1 PRELIMINARY: FEDERATED LEARNING

We assume a standard federated learning setup with a server that can transmit and receive messages from $M$ client devices. Each client $m \in [M]$ has access to $N_m$ training instances $\{(x_i, y_i)\}_{i=1}^{N_m}$ in the form of image $x_i \in \mathcal{X}$ and corresponding labels $y_i \in \mathcal{Y}$ that are drawn i.i.d. from a device-indexed joint distribution, *i.e.*, $(x_i, y_i) \sim \mathbb{P}_m(x, y)$. The goal of standard federated learning is to learn a deep neural network: $f(\boldsymbol{w}_g, \boldsymbol{w}_h) \triangleq g(\boldsymbol{w}_g) \circ h(\boldsymbol{w}_h)$, where $h : \mathcal{X} \to \mathcal{Z}$ is a feature extractor with $K$ convolutional stages: $h = h^K \circ h^{K-1} \circ \cdots \circ h^1$, and $g : \mathcal{Z} \to \mathcal{Y}$ is a classifier. To learn network parameters $\boldsymbol{w} = \{\boldsymbol{w}_g, \boldsymbol{w}_h\}$, the empirical risk minimization (ERM) is widely used:

$$\mathcal{L}^{\text{ERM}}(\boldsymbol{w}) \triangleq \frac{1}{M} \sum_{m \in [M]} \mathcal{L}_m^{\text{ERM}}(\boldsymbol{w}), \quad \text{where } \mathcal{L}_m^{\text{ERM}}(\boldsymbol{w}) = \mathbb{E}_{(x_i, y_i) \sim \mathbb{P}_m}[\ell_i(g \circ h(x_i), y_i; \boldsymbol{w})]. \quad (1)$$

Here the global objective $\mathcal{L}^{\text{ERM}}$ is decomposable as a sum of device-level empirical loss objectives (*i.e.*, $\{\mathcal{L}_m^{\text{ERM}}\}_m$). Each $\mathcal{L}_m^{\text{ERM}}$ is computed based on a per-data loss function $\ell_i$. Due to the separation of clients' data, $\mathcal{L}^{\text{ERM}}(\boldsymbol{w})$ cannot be solved directly. FEDAVG (McMahan et al., 2017) is a leading algorithm to address this. It starts with *client training* of all the clients in parallel, with each client optimizing $\mathcal{L}_m^{\text{ERM}}$ independently. After local client training, FEDAVG performs *model aggregation* to average all client models into a updated global model, which will be distributed back to the clients for the next round of client training. Here the *client training* objective in FEDAVG is equivalent to empirically approximating the local distribution $\mathbb{P}_m$ by a finite $N_m$ number of examples, *i.e.*, $\mathbb{P}_m^e(x, y) = 1/N_m \sum_{i=1}^{N_m} \delta(x = x_i, y = y_i)$, where $\delta(x = x_i, y = y_i)$ is a Dirac mass centered at $(x_i, y_i)$.

## 2.2 MOTIVATION

While the ERM-based formulation has achieved great success, it is straightforward to see that the solution would strongly depend on how each approximated local distribution $\mathbb{P}_m^e$ mimics the underlying universal distribution $\mathbb{P}$. In real-world federated learning setup however, in all but trivial cases each $\mathbb{P}_m^e$ exhibits a unique distribution shift from $\mathbb{P}$, which causes not only inconsistency between local and global empirical losses (Acar et al., 2021; Wang et al., 2020), but also generalization issues (Yuan et al., 2022). In this work, we circumvent this issue by fitting each local dataset a *richer* distribution (instead of the delta distribution) in the *vicinal region* of each sample $(x_i, y_i)$ so as to estimate a more informed risk. This is precisely the principle behind vicinal risk minimization (VRM) (Chapelle et al., 2000). Particularly, for data point $(x_i, y_i)$, a vicinity distribution $\mathbb{V}_m(\hat{x}_i, \hat{y}_i | x_i, y_i)$ is defined, from which novel virtual samples can be generated to enlarge the support of the local data distribution. In this way, we obtain an improved approximation of $\mathbb{P}_m$ as $\mathbb{P}_m^v = 1/N_m \sum_{i=1}^{N_m} \mathbb{V}_m(\hat{x}_i, \hat{y}_i | x_i, y_i)$. In centralized learning scenarios, various successful instances of $\mathbb{V}_m$, *e.g.*, MIXUP (Zhang et al., 2018), CUTMIX (Yun et al., 2019), have been developed. Simply applying them to local clients, though allowing for performance improvements (see Table 6), is sub-optimal since, without injecting any global information, $\mathbb{P}_m^v$ only provides a better approximation to the local distribution $\mathbb{P}_m$, rather than the true distribution $\mathbb{P}$. We solve this by introducing a dedicated method FEDFA to estimate more reasonable $\mathbb{V}_m$ in federated learning.

## 2.3 FEDFA: FEDERATED FEATURE AUGMENTATION

FEDFA belongs to the family of label-preserving feature augmentation (Xie et al., 2020). During training, it estimates a vicinity distribution $\mathbb{V}_m^k$ at each layer $h^k$ to augment hidden features in client $m$. Considering $\boldsymbol{X}_m^k \in \mathbb{R}^{B \times C \times H \times W}$ as the intermediate feature representation of $B$ mini-batch images, with spatial size $(H \times W)$ and channel number $(C)$, and $Y_m^k$ as corresponding label. $\mathbb{V}_m^k$ is label-preserving in the sense that $\mathbb{V}_m^k(\hat{\boldsymbol{X}}_m^k, \hat{Y}_m | \boldsymbol{X}_m^k, Y_m) \triangleq \mathbb{V}_m^k(\hat{\boldsymbol{X}}_m^k | \boldsymbol{X}_m^k)\delta(\hat{Y}_m = Y_m)$, *i.e.*, it only transforms the latent feature $\boldsymbol{X}_m^k$ to $\hat{\boldsymbol{X}}_m^k$, but preserves the original label $Y_m^k$.

### 2.3.1 FEDERATED FEATURE AUGMENTATION FROM A PROBABILISTIC VIEW

Instead of explicitly modeling $\mathbb{V}_m^k(\hat{\boldsymbol{X}}_m^k | \boldsymbol{X}_m^k)$, our method performs implicit feature augmentation by manipulating channel-wise feature statistics. Specifically, for $\boldsymbol{X}_m^k$, its channel-wise statistics, *i.e.*, mean $\mu_m^k$ and standard deviation $\sigma_m^k$ are given as follows:

$$\mu_m^k = \frac{1}{HW} \sum_{h=1}^{H} \sum_{w=1}^{W} \boldsymbol{X}_m^{k,(h,w)} \ \in \mathbb{R}^{B \times C}, \quad \sigma_m^k = \sqrt{\frac{1}{HW} \sum_{h=1}^{H} \sum_{w=1}^{W} (\boldsymbol{X}_m^{k,(h,w)} - \mu_m^k)^2} \ \in \mathbb{R}^{B \times C}, \quad (2)$$

where $\boldsymbol{X}_m^{k,(h,w)} \in \mathbb{R}^{B \times C}$ represents features at spatial location $(h, w)$. As the abstract of latent features, these statistics carry domain-specific information (*e.g.*, style). They are instrumental to image generation (Huang & Belongie, 2017), and have been recently used for data augmentation in image recognition (Li et al., 2021a). In heterogeneous federated learning scenarios, the feature statistics among local clients will be inconsistent, and exhibit uncertain feature statistic shifts from the statistics of the true distribution. Our method explicitly captures such shift via probabilistic modeling. Concretely, instead of representing each feature $\boldsymbol{X}_m^k$ with deterministic statistics $\{\mu_m^k, \sigma_m^k\}$, we hypothesize that the feature is conditioned on probabilistic statistics $\{\hat{\mu}_m^k, \hat{\sigma}_m^k\}$, which are sampled around the original statistics based on a multi-variate Gaussian distribution, *i.e.*, $\hat{\mu}_m^k \sim \mathcal{N}(\mu_m^k, \hat{\Sigma}_{\mu_m^k}^2)$ and $\hat{\sigma}_m^k \sim \mathcal{N}(\sigma_m^k, \hat{\Sigma}_{\sigma_m^k}^2)$, where each Gaussian's center corresponds to the original statistic, and the variance is expected to capture the potential feature statistic shift from the true distribution. Our core goal is thus to estimate proper variances $\hat{\Sigma}_{\mu_m^k}^2 / \hat{\Sigma}_{\sigma_m^k}^2$ for reasonable and informative augmentation.

**Client-specific Statistic Variances.** In *client-side*, we compute client-specific variances of feature statistics based on the information within each mini-batch:

$$\Sigma_{\mu_m^k}^2 = \frac{1}{B} \sum_{b=1}^{B} (\mu_m^k - \mathbb{E}[\mu_m^k])^2 \ \in \mathbb{R}^C, \quad \Sigma_{\sigma_m^k}^2 = \frac{1}{B} \sum_{b=1}^{B} (\sigma_m^k - \mathbb{E}[\sigma_m^k])^2 \ \in \mathbb{R}^C, \quad (3)$$

where $\Sigma_{\mu_m^k}^2$ and $\Sigma_{\sigma_m^k}^2$ denote the variance of feature mean $\mu_m^k$ and standard deviation $\sigma_m^k$ that are specific to each client. Each value in $\Sigma_{\mu_m^k}^2$ or $\Sigma_{\sigma_m^k}^2$ is the variance of feature statistics in a particular channel, and its magnitude manifests how the channel will change potentially in the feature space.

**Client-sharing Statistic Variances.** The client-specific variances are solely computed based on the data in each individual client, and thus likely biased due to local dataset bias. To solve this, we further estimate client-sharing feature statistic variances taking information of all clients into account. Particularly, we maintain a momentum version of feature statistics for each client, which are online estimated during training:

$$\bar{\mu}_m^k \leftarrow \alpha \bar{\mu}_m^k + (1-\alpha)\frac{1}{B}\sum_{b=1}^{B}\mu_m^k \ \in \mathbb{R}^C, \quad \bar{\sigma}_m^k \leftarrow \alpha \bar{\sigma}_m^k + (1-\alpha)\frac{1}{B}\sum_{b=1}^{B}\sigma_m^k \ \in \mathbb{R}^C, \tag{4}$$

where $\bar{\mu}_m^k$ and $\bar{\sigma}_m^k$ are the momentum updated feature statistics of layer $h^k$ in client $m$, and they are initialized as $C$-dimensional all-zero and all-one vectors, respectively. $\alpha$ is a momentum coefficient. We set a same $\alpha$ for both updating, and found no benefit to set it differently. In each communication, these accumulated local feature statistics are sent to the server along with model parameters. Let $\bar{\mu}^k = [\bar{\mu}_1^k, \ldots, \bar{\mu}_M^k] \in \mathbb{R}^{M \times C}$ and $\bar{\sigma}^k = [\bar{\sigma}_1^k, \ldots, \bar{\sigma}_M^k] \in \mathbb{R}^{M \times C}$ denote collections of accumulated feature statistics of all clients, the client sharing statistic variances are determined in *server-side* by:

$$\Sigma_{\mu^k}^2 = \frac{1}{M}\sum_{m=1}^{M}(\bar{\mu}_m^k - \mathbb{E}[\bar{\mu}^k])^2 \ \in \mathbb{R}^C, \quad \Sigma_{\sigma^k}^2 = \frac{1}{M}\sum_{m=1}^{M}(\bar{\sigma}_m^k - \mathbb{E}[\bar{\sigma}^k])^2 \ \in \mathbb{R}^C. \tag{5}$$

In addition, it is intuitive that some channels are more potentially to change than others, and it will be favorable to highlight these channels to enable a sufficient and reasonable exploration of the space of feature statistics. To this end, we further modulate client sharing estimations with a *Student's t-distribution* (Student, 1908; Van der Maaten & Hinton, 2008) with one degree of freedom to convert the variances to probabilities. The *t-distribution* has heavier tails than other alternatives such as Gaussian distribution, allowing to highlight the channels with larger statistic variance, at the same time, avoiding overly penalizing the others. Formally, denote $\Sigma_{\mu^k}^{2,(j)}$ and $\Sigma_{\sigma^k}^{2,(j)}$ as the shared variances of the $j$th channel in $\Sigma_{\mu^k}^2$ and $\Sigma_{\sigma^k}^2$ (Eq. 5), respectively. They are modulated by the *t-distribution* as follows:

$$\gamma_{\mu^k}^{(j)} = \frac{C(1 + {}^1\!/_{\Sigma_{\mu^k}^{2,(j)}})^{-1}}{\sum_{c=1}^{C}(1 + {}^1\!/_{\Sigma_{\mu^k}^{2,(c)}})^{-1}} \ \in \mathbb{R}, \quad \gamma_{\sigma^k}^{(j)} = \frac{C(1 + {}^1\!/_{\Sigma_{\sigma^k}^{2,(j)}})^{-1}}{\sum_{c=1}^{C}(1 + {}^1\!/_{\Sigma_{\sigma^k}^{2,(c)}})^{-1}} \ \in \mathbb{R}, \tag{6}$$

where $\gamma_{\mu^k}^{(j)}$ and $\gamma_{\sigma^k}^{(j)}$ refer to the modulated variances of the $j$-th channel. By applying Eq. 6 to each channel separately, we obtain $\gamma_{\mu^k} = [\gamma_{\mu^k}^{(1)}, \ldots, \gamma_{\mu^k}^{(C)}] \in \mathbb{R}^C$ and $\gamma_{\sigma^k} = [\gamma_{\sigma^k}^{(1)}, \ldots, \gamma_{\sigma^k}^{(C)}] \in \mathbb{R}^C$ as modulated statistic variances of all feature channels at layer $h^k$. In this way, the channels with large values in $\Sigma_{\mu^k}^2$ (or $\Sigma_{\sigma^k}^2$) will be assigned with much higher importance in $\gamma_{\mu^k}$ (or $\gamma_{\sigma^k}$) than other channels, allowing for more extensive augmentation along those directions.

**Adaptive Variance Fusion.** The modulated client sharing estimations $\{\gamma_{\mu^k}, \gamma_{\sigma^k}\}$ provide a quantification of distribution difference among clients, and larger values imply potentials of more significant changes of corresponding channels in the true feature statistic space. Therefore, for each client, we weight the client specific statistic variances $\{\Sigma_{\mu_m^k}^2, \Sigma_{\sigma_m^k}^2\}$ by $\{\gamma_{\mu^k}, \gamma_{\sigma^k}\}$, so that each client has a sense of such difference. To avoid overly modification of client specific statistic variances, we add a residual connection for fusion, yielding an estimation of Gaussian ranges as:

$$\hat{\Sigma}_{\mu_m^k}^2 = (\gamma_{\mu^k} + 1) \odot \Sigma_{\mu_m^k}^2 \ \in \mathbb{R}^C, \quad \hat{\Sigma}_{\sigma_m^k}^2 = (\gamma_{\sigma^k} + 1) \odot \Sigma_{\sigma_m^k}^2 \ \in \mathbb{R}^C, \tag{7}$$

where $\odot$ denotes the Hadamard product.

**Implementation of Feature Augmentation.** After establishing the Gaussian distribution, we synthesize novel feature $\hat{X}_m^k$ in the vicinity of $X_m^k$ as follows:

$$\hat{X}_m^k = \hat{\sigma}_m^k \frac{X_m^k - \mu_m^k}{\sigma_m^k} + \hat{\mu}_m^k, \quad \text{where} \ \ \hat{\mu}_m^k \sim \mathcal{N}(\mu_m^k, \hat{\Sigma}_{\mu_m^k}^2), \ \ \hat{\sigma}_m^k \sim \mathcal{N}(\sigma_m^k, \hat{\Sigma}_{\sigma_m^k}^2). \tag{8}$$

Here $X_m^k$ is first normalized with its original statistics by ${(X_m^k - \mu_m^k)}/{\sigma_m^k}$, and further scaled with novel statistics $\{\hat{\mu}_m^k, \hat{\sigma}_m^k\}$ that are randomly sampled from corresponding Gaussian distribution. To make the sampling differentiable, we use the re-parameterization trick (Kingma & Welling, 2013):

$$\hat{\mu}_m^k = \mu_m^k + \epsilon_\mu \hat{\Sigma}_{\mu_m^k}, \quad \hat{\sigma}_m^k = \sigma_m^k + \epsilon_\sigma \hat{\Sigma}_{\sigma_m^k}, \tag{9}$$

where $\epsilon_\mu \sim \mathcal{N}(0,1)$ and $\epsilon_\sigma \sim \mathcal{N}(0,1)$ follow the normal Gaussian distribution.

The proposed federated feature augmentation (FFA) operation in Eq. 8 is a plug-and-play layer, *i.e.*, it can be inserted at arbitrary layers in the feature extractor $h$. In our implementation, we add a FFA layer after each convolutional stage of the networks. During training, we follow the stochastic learning strategy (Verma et al., 2019; Zhou et al., 2021; Li et al., 2022b) to activate each FFA layer with a probability of $p$. This allows for more diverse augmentation from iteration to iteration (based on the activated FFA layers). At test time, no augmentation is applied. In Appendix A, we provide detailed descriptions of FEDFA in Algorithm 1 and FFA in Algorithm 2.

## 3 THEORETICAL INSIGHTS

In this section, we provide mathematical analysis to gain deeper insights into FEDFA. To begin with, we show that FEDFA is a noise injection process (Bishop, 1995; Camuto et al., 2020; Lim et al., 2022) that injects federation-aware noises to latent features.

**Lemma 1.** *Consider client $m \in [M]$, for a batch-wise latent feature $\boldsymbol{X}_m^k$ at layer $k$, its augmentation in* FEDFA *(cf. Eq. 8) follows a noising process $\hat{\boldsymbol{X}}_m^k = \boldsymbol{X}_m^k + \boldsymbol{e}_m^k$, with the noise $\boldsymbol{e}_m^k$ taking the form:*

$$\boldsymbol{e}_m^k = \epsilon_\sigma \hat{\Sigma}_{\sigma_m^k} \bar{\boldsymbol{X}}_m^k + \epsilon_\mu \hat{\Sigma}_{\mu_m^k}, \tag{10}$$

*where $\epsilon_\mu \sim \mathcal{N}(0,1)$, $\epsilon_\sigma \sim \mathcal{N}(0,1)$, $\bar{\boldsymbol{X}}_m^k = (\boldsymbol{X}_m^k - \mu_m^k)/\sigma_m^k$.*

Based on Lemma 1, we can identify the federation-aware implicit regularization effects of FEDFA.

**Theorem 1.** *In* FEDFA*, the loss function $\mathcal{L}_m^{\text{FEDFA}}$ of client $m$ can be expressed as:*

$$\mathcal{L}_m^{\text{FEDFA}} = \mathcal{L}_m^{ERM} + \mathcal{L}_m^{REG}, \tag{11}$$

*where $\mathcal{L}_m^{ERM}$ is the standard ERM loss, and $\mathcal{L}_m^{REG}$ is the regularization term:*

$$\mathcal{L}_m^{ERM} = \mathbb{E}_{(X_m, Y_m) \sim \mathbb{P}_m} \ell(g(h^{1:K}(X_m)), Y_m), \tag{12}$$

$$\mathcal{L}_m^{REG} = \mathbb{E}_{\mathcal{Z} \sim \mathcal{K}} \mathbb{E}_{(X_m, Y_m) \sim \mathbb{P}_m} \nabla_{h^{1:K}(X_m)} \ell(g(h^{1:K}(X_m)), Y_m)^\top \sum_{z \in \mathcal{Z}} \boldsymbol{J}^z(X_m) \boldsymbol{e}_m^z, \tag{13}$$

*where $\boldsymbol{J}^z$ denotes the Jacobian of layer $z$ (see Proposition 1 in Appendix for its explicit expression).*

Theorem 1 implies that, FEDFA implicitly introduces regularization to local client learning by regularizing the gradients of latent representations (*i.e.*, $\nabla_{h^{1:K}(X_m)} \ell(g(h^{1:K}(X_m)), Y_m)^\top$), weighted by federation-aware noises in Lemma 1, *i.e.*, $\sum_{z \in \mathcal{Z}} \boldsymbol{J}^z(X_m) \boldsymbol{e}_m^z$.

## 4 EMPIRICAL RESULT

### 4.1 SETUP

**Datasets.** We conduct extensive experiments on five datasets: Office-Caltech 10 (Gong et al., 2012), DomainNet(Peng et al., 2019) and ProstateMRI(Liu et al., 2020b) for validation of FEDFA in terms of feature-shift non-IID, as well as larger-scale datasets CIFAR-10 (Krizhevsky & Hinton, 2009) and EMNIST (Cohen et al., 2017) for cases of label distribution and data size heterogeneity, respectively.

**Baselines.** For comprehensive evaluation, we compare FEDFA against several state-of-the-art federated learning techniques, including FEDAVG (McMahan et al., 2017), FEDAVGM (Hsu et al., 2019), FEDPROX (Li et al., 2020a), FEDSAM (Qu et al., 2022), FEDBN (Li et al., 2020b), FEDROBUST (Reisizadeh et al., 2020), and FEDMIX (Yoon et al., 2021). Moreover, we compare with FEDHARMO (Jiang et al., 2022) in ProstateMRI, that is specialized designed for medical imaging.

To gain more insights into FEDFA, we develop two baselines: FEDFA-R(ANDOM) and FEDFA-C(LIENT). FEDFA-R randomly perturbs feature statistics based on Gaussian distribution with a same standard deviation for all channels, *i.e.*, $\hat{\Sigma}_{\mu_m^k} = \hat{\Sigma}_{\sigma_m^k} = \lambda$, where $\lambda = 0.5$. FEDFA-C performs augmentation based only on client specific variances, *i.e.*, Eq. 7 turns into $\hat{\Sigma}_{\mu_m^k}^2 = \Sigma_{\mu_m^k}^2$, $\hat{\Sigma}_{\sigma_m^k}^2 = \Sigma_{\sigma_m^k}^2$.

**Metrics.** As conventions, we use top-1 accuracy for image classification and Dice coefficient for medical image segmentation, respectively. We report the performance only for the global model.

**Implementation Details.** We use PyTorch to implement FEDFA and other baselines. Following FEDBN (Li et al., 2020b), we adopt AlexNet (Krizhevsky et al., 2017) on Office-Caltech 10 and

Table 1: **Image classification performance on Office-Caltech 10 and DomainNet `test`.** Top-1 accuracy (%) is reported. Office-Caltech 10 has four clients: A(mazon), C(altech), D(SLR), and W(ebcam), while DomainNet has six: C(lipart), I(nfograph), P(ainting), Q(uickdraw), R(eal), and S(ketch). See §4.2 for details.

| Algorithm | Office-Caltech 10 (Gong et al., 2012) | | | | | DomainNet (Peng et al., 2019) | | | | | | |
|---|---|---|---|---|---|---|---|---|---|---|---|---|
| | A | C | D | W | Average | C | I | P | Q | R | S | Average |
| FEDAVG | 84.4 | 66.7 | 75.0 | 88.1 | 78.5 | 71.5 | 33.2 | 57.8 | 76.5 | 72.9 | 65.2 | 62.8 |
| FEDPROX | 84.9 | 64.0 | 78.1 | 88.1 | 78.8 | 70.9 | 32.9 | 61.2 | 74.1 | 71.1 | 67.9 | 63.0 |
| FEDSAM | 81.7 | 63.1 | 50.0 | 81.4 | 69.1 | 60.1 | 30.1 | 53.0 | 64.8 | 61.9 | 47.3 | 52.9 |
| FEDAVGM | 85.9 | 64.0 | 71.9 | 94.9 | 79.2 | 79.8 | 33.3 | 58.8 | 72.6 | 72.8 | 66.1 | 62.5 |
| FEDROBUST | 82.3 | 64.0 | 81.3 | 93.2 | 80.2 | 70.9 | 32.9 | 60.7 | 75.7 | 72.6 | 68.5 | 63.6 |
| FEDBN | 82.3 | 63.6 | 81.2 | 94.9 | 80.5 | 72.4 | 32.7 | 64.3 | 74.0 | 69.9 | 70.8 | 64.0 |
| FEDMIX | 81.7 | 63.1 | 81.3 | 93.2 | 79.8 | 75.9 | 34.1 | 61.7 | 73.8 | 69.4 | 70.6 | 64.3 |
| **FEDFA** | 88.0 | 65.8 | 90.6 | 88.1 | **83.1** | 77.4 | 34.9 | 61.2 | 78.8 | 73.2 | 73.5 | **66.5** |

DomainNet, using the SGD optimizer with learning rate 0.01 and batch size 32. Following FED-HARMO (Jiang et al., 2022), we employ U-Net (Ronneberger et al., 2015) on ProstateMRI using Adam as the optimizer with learning rate 1e-4 and batch size 16. The communication rounds are 400 for Office-Caltech 10 and DomainNet, and 500 for ProstateMRI, with the number of local update epoch setting to 1 in all cases. For EMNIST, we strictly follow FEDMIX (Yoon et al., 2021) to introduce data size heterogeneity by partitioning data w.r.t. writers, and train a LeNet-5 (LeCun et al., 1998) using SGD with batch size 10. The total number of clients is 200 and only 10 clients are sampled per communication round. We run 200 rounds in total. For CIFAR-10, we sample local data based on Dirichlet distribution $Dir(\alpha)$ to simulate label distribution heterogeneity. As (Qu et al., 2022; Kim et al., 2022), we set $\alpha$ to 0.3 or 0.6, and train a ResNet-18 (He et al., 2016). The number of clients is 100 with participation rate 0.1, while the number of communication round is set to 100.

## 4.2 MAIN RESULTS

We first present the overall results on the five benchmarks, *i.e.*, Office-Caltech 10 and DomainNet in Table 1 and Fig. 1, ProstateMRI in Table 2 and Fig. 2, EMNIST and CIFAR-10 in Table 3.

**Results on Office-Caltech 10 and DomainNet.** FEDFA *yields solid improvements over competing methods for image classification.* As presented in Table 1, FEDFA leads to consistent performance gains over the competitors across the benchmarks. The improvements over FE-DAVG can be as large as **4.6%** and **3.7%** on Office-Caltech 10 and DomainNet, respectively. Moreover, in comparison to prior data augmentation-based algorithm FEDMIX, FEDFA also brings solid gains, *i.e.*, **3.3%** on Office-Caltech 10 and **2.2%** on DomainNet. This is encouraging since our approach in nature better preserves privacy than FEDMIX, which requires exchanging aver-

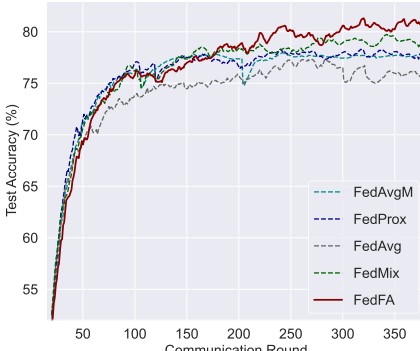

Figure 1: **Test accuracy versus communication rounds** on Office-Caltech 10.

aged data across clients. Moreover, Fig. 1 depicts the convergence curves of comparative methods on Office-Caltech 10. At the early training stage ($0 \sim 200$ rounds), FEDFA shows similar training efficiency as other baselines like FEDPROX and FEDMIX. But as the training goes, FEDFA is able to converge to a more optimal solution.

**Results on ProstateMRI.** FEDFA *shows leading performance with extremely small local datasets.* In some practical scenarios like healthcare, the size of local dataset can be very small, which poses a challenge for federated learning. To examine the performance of federated learning algorithms in this scenario, we build *mini*-ProstateMRI by randomly sampling only $^1/_6$ of all training samples in each client for training. Results are summarized in Table 2. FEDFA outperforms FEDAVG by significant margins (*i.e.*, **3.0%**) and it even performs better than FEDHARMO, which is specifically designed for medical scenarios. In addition, Fig. 2 shows how the performance of methods varies with respect to the size of local dataset. We train methods with different fractions (*i.e.*, $^1/_6$, $^2/_6$, $^3/_6$, $^4/_6$, $^5/_6$, 1) of training samples. FEDFA shows promising performance in all cases.

Table 2: **Medical image segmentation accuracy on *mini-ProstateMRI* `test`** (Liu et al., 2020b) with small-size local datasets. Dice score (%) is reported. The dataset consists of data from six medical institutions: B(IDMC), H(K), I(2CVB), (B)M(C), R(UNMC) and U(CL). The number in the bracket denotes the number of training samples in each client. See §4.2 for more details.

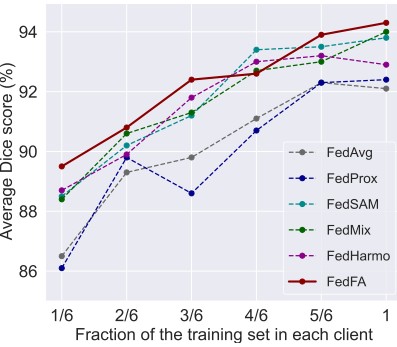

| Algorithm | B (32) | H (32) | I (46) | M (38) | R (41) | U (32) | Average |
|---|---|---|---|---|---|---|---|
| FEDAVG | 81.2 | 90.8 | 86.1 | 84.0 | 91.0 | 86.2 | 86.5 |
| FEDPROX | 82.8 | 89.1 | 89.8 | 79.5 | 89.8 | 85.6 | 86.1 |
| FEDAVGM | 80.3 | 91.6 | 88.2 | 82.2 | 91.2 | 86.5 | 86.7 |
| FEDSAM | 82.7 | 92.5 | 91.8 | 83.6 | 92.6 | 88.1 | 88.5 |
| FEDROBUST | 81.7 | 91.3 | 91.5 | 88.5 | 89.4 | 84.2 | 87.7 |
| FEDBN | 88.9 | 92.3 | 90.6 | 88.1 | 87.6 | 85.4 | 88.8 |
| FEDMIX | 86.3 | 91.6 | 89.6 | 88.1 | 89.8 | 85.2 | 88.4 |
| FEDHARMO | 86.7 | 91.6 | 92.7 | 84.2 | 92.5 | 84.6 | 88.7 |
| **FEDFA** | 85.7 | 92.6 | 91.0 | 85.4 | 92.9 | 89.2 | **89.5** |

Figure 2: **Segmentation performance** w.r.t local data size (*i.e.*, fraction of training samples over the whole training set).

**Results on CIFAR-10 and EMNIST.** *In addition to feature-shift non-i.i.d.,* FEDFA *shows consistent improvements in label distribution heterogeneity (CIFAR-10) and data size heterogeneity (EMNIST).* As shown in Table 3, in CIFAR-10, FEDFA surpasses the second best method, FEDMIX, by **0.8%** and **1.2%** with respect to two non-i.i.d levels Dir(0.6) and Dir(0.3), respectively. Notably, as the non-i.i.d. level increasing from Dir(0.6) to Dir(0.3), FEDFA tends to yield a larger gap of performance gain, showing a strong capability in handling severe non-i.i.d. scenarios.

Table 3: **Performance on CIFAR-10 and EMNIST.**

| Algorithm | CIFAR-10 | | EMNIST |
|---|---|---|---|
| | Dir (0.6) | Dir (0.3) | |
| FEDAVG | 73.3 | 69.2 | 84.9 |
| FEDAVGM | 73.4 | 69.1 | 85.5 |
| FEDPROX | 74.0 | 69.5 | 84.9 |
| FEDBN | 73.7 | 69.8 | 85.3 |
| FEDSAM | 74.3 | 70.0 | 86.5 |
| FEDROBUST | 74.9 | 70.5 | 86.7 |
| FEDMIX | 75.5 | 70.7 | 86.6 |
| **FEDFA** | **76.3** | **71.9** | **87.8** |

In addition, in EMNIST, FEDFA outperforms FEDROBUST by **1.1%** and FEDMIX by **1.2%**, respectively. These results reveal that though designed for feature shift non-i.i.d., FEDFA's very nature of data augmentation makes it a fundamental technique to various non-i.i.d. challenges in FL.

Table 4: **Comparison of generalization performance to unseen test clients** on the three benchmarks (§4.2).

| Algorithm | Office-Caltech 10 | | | | | DomainNet | | | | | | | ProstateMRI | | | | | | |
|---|---|---|---|---|---|---|---|---|---|---|---|---|---|---|---|---|---|---|---|
| | A | C | D | W | Avg | C | I | P | Q | R | S | Avg | B | H | I | M | R | U | Avg |
| FEDAVG | 64.6 | 49.3 | 71.9 | 55.9 | 60.4 | 63.1 | 27.5 | 49.6 | 44.7 | 51.7 | 48.2 | 47.5 | 60.7 | 85.3 | 78.4 | 67.2 | 83.0 | 59.0 | 72.3 |
| FEDPROX | 63.0 | 50.7 | 68.7 | **62.7** | 61.3 | 62.2 | 26.9 | 49.6 | 42.4 | 50.5 | 48.9 | 46.8 | 61.5 | 86.2 | **79.3** | 68.6 | 84.5 | 62.4 | 73.8 |
| FEDROBUST | 64.9 | 53.0 | 73.2 | 58.1 | 62.3 | 63.5 | 28.5 | 49.8 | 44.6 | 53.5 | 56.7 | 49.4 | 62.4 | 87.2 | 78.0 | 77.1 | 88.0 | 65.3 | 76.3 |
| FEDMIX | 65.1 | 52.6 | 73.8 | 58.9 | 62.6 | 63.3 | 28.0 | **50.1** | 45.9 | 53.3 | 56.8 | 49.6 | 62.1 | 86.7 | 78.1 | 76.8 | 87.7 | 65.6 | 76.2 |
| **FEDFA** | **65.6** | **54.2** | **78.1** | 59.3 | **64.3** | **64.1** | **28.8** | 49.4 | **47.5** | **56.6** | **61.0** | **51.2** | **64.0** | **88.3** | 75.9 | **79.0** | **89.1** | **68.8** | **77.5** |

## 4.3 FEDERATED DOMAIN GENERALIZATION PERFORMANCE

Federated learning are dynamic systems, in which novel clients may enter the system after model training, most possibly with test-time distribution shift. However, most prior federated learning algorithms focus only on improving model performance on the participating clients, while neglecting model generalizability to unseen non-participating clients. Distribution shift often occurs during deployment, thus it is essential to evaluate the generalizability of federated learning algorithms. With a core of data augmentation, FEDFA is supposed to enforce regularization to neural network learning, which could improve generalization capability.

To verify this, we perform experiments for federated domain generalization based on the *leave-one-client-out* strategy, *i.e.*, training on $M-1$ distributed clients and testing on the held-out unparticipating client. The results are presented in Table 4. As seen, FEDFA achieves leading generalization performance on most unseen clients. For example, it yields consistent improvements as compared to FEDMIX, *i.e.*, **1.7%** on Office-Caltech 10, **1.6%** on DomainNet, and **1.3%** on ProstateMRI, in terms of average performance. Despite the improved performance, we find by comparing to the results reported in Table 1 and Table 2 that, current federated learning algorithms still encounter significant participation gap (Yuan et al., 2022), *i.e.*, the performance difference between participating and non-participating clients, which is a critical issue that should be tackled in future.

### 4.4 Diagnostic Experiment

We conduct a set of ablative experiments to enable a deep understanding of FEDFA.

**FEDFA vs. FEDFA-C and FEDFA-R.** We first verify FEDFA against the two baseline variants mentioned in §4.1. Both variants involve only device-dependent augmentation of feature statistics, without explicitly considering any global information. As shown in Table 5, by randomly perturbing

Table 5: Efficacy of FEDFA over FEDFA-C and FEDFA-R.

| Variant | Office | DomainNet | ProstateMRI |
|---|---|---|---|
| FEDAVG | 78.5 | 62.8 | 86.5 |
| FEDFA-R | 78.6 | 61.0 | 86.1 |
| FEDFA-C | 79.5 | 63.7 | 87.8 |
| **FEDFA** | **83.1** | **66.5** | **89.5** |

feature statistics, FEDFA-R shows no improvements or even suffers performance degradation on DomainNet and ProstateMRI against FEDAVG; FEDFA-C yields promising performance gains by taking into account client-specific feature statistic variances; by comparing FEDFA and FEDFA-C, we confirm the significance of universal feature statistic information in federated augmentation.

**FEDFA vs. traditional augmentation methods.** We compare FEDFA with four conventional data/feature augmentation techniques, *i.e.*, MIXUP (Zhang et al., 2018), MANIFOLD MIXUP (Verma et al., 2019), MIXSTYLE (Zhou et al., 2021) and MoEX (Li et al., 2021a). The results are presented in Table 6. We show that i) all the four techniques yield non-trivial improvements over FEDAVG, and some of them (*e.g.*, MoEX) even outperform well-

Table 6: Efficacy of FEDFA against augmentation techniques.

| Algorithm | Office | DomainNet | ProstateMRI |
|---|---|---|---|
| FEDAVG | 78.5 | 62.8 | 86.5 |
| MIXUP | 79.2 | 63.4 | 87.0 |
| M-MIXUP | 79.6 | 63.5 | 87.6 |
| MIXSTYLE | 79.9 | 64.1 | 88.5 |
| MoEX | 80.2 | 64.6 | 88.3 |
| **FEDFA** | **83.1** | **66.5** | **89.5** |
| **FEDFA**+MIXUP | 83.7 | 67.0 | 89.9 |
| **FEDFA**+M-MIXUP | 83.6 | 66.9 | 90.2 |
| **FEDFA**+MIXSTYLE | 84.0 | 67.2 | 90.2 |
| **FEDFA**+MoEX | 83.9 | 67.0 | 90.1 |

designed federated learning algorithms (as compared to Tables 1-2); by accounting for global feature statistics, FEDFA surpasses all of them, yielding **2.9%/1.9%/1.0%** improvements over the second-best results on Office/DomainNet/ProstateMRI, respectively; iii) combining FEDFA with these techniques allows further performance uplifting, verifying the complementary roles of FEDFA to them.

**Adaptive Variance Fusion.** Next, we examine the effect of adaptive variance fusion in Eqs. 6-7. We design a baseline "Direct Fusion" that directly combines the client-specific and client-sharing statistic

Table 7: Effectiveness of adaptive variance fusion.

| Variant | Office | DomainNet | ProstateMRI |
|---|---|---|---|
| Direct Fusion | 80.6 | 64.1 | 86.9 |
| Adaptive Fusion | 83.1 | 66.5 | 89.5 |

variances as: $\hat{\Sigma}^2_{\mu^k_m} = (\Sigma^2_{\mu^k}+1)\Sigma^2_{\mu^k_m}, \hat{\Sigma}^2_{\sigma^k_m} = (\Sigma^2_{\sigma^k}+1)\Sigma^2_{\sigma^k_m}$. We find from Table 7 that the baseline encounters severe performance degradation across all three benchmarks. A possible reason is that the two types of variances are mostly mis-matched, and the simple fusion strategy may cause significant changes of client-specific statistic variances, which would be harmful for local model learning.

**Hyper-parameter analysis.** FEDFA includes only two hyper-parameters, *i.e.*, momentum coefficient $\alpha$ in Eq. 4 and stochastic learning probability $p$ to apply feature statistic augmentation during training. As shown in Fig. 3, (1) the model is overall robust to $\alpha$. Notably, it yields promising performance at $\alpha=0$, in which the model only uses the feature statistics of the last mini-batch in each local epoch to compute client-sharing statistic variances. This result reveals that FEDFA is insensitive to errors of client-sharing statistic variances. (2) For the probability $p$, we see that FEDFA significantly improves the baseline (*i.e.*, $p=0$), even with a small probability (*e.g.*, $p=0.1$). The best performance is reached at $p=0.5$.

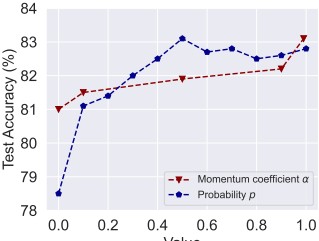

Figure 3: **Hyper-parameter analysis for $\alpha$ and $p$.**

### 4.5 Complexity Analysis

**Computation and memory costs.** FEDFA involves only several basic matrix operations, thus incurring negligible extra computation cost. Compared to FEDAVG, it requires $4\sum_{k=1}^{K} C_k$ more GPU memory allocation to store four statistic values ($\bar{\mu}^k_m, \bar{\sigma}^k_m, \gamma_{\mu^k}, \gamma_{\sigma^k}$) at each of the $K$ FFA layers. Here $C_k$ is the number of feature channel at each layer $k$. The costs are in practice very minor, *e.g.*, 18 KB/15.5 KB for AlexNet/U-Net. For comparison, FEDMIX requires $2\times$ more GPU memory than FEDAVG. The low computation/memory costs make FEDFA favorable for edge devices.

**Communication cost.** In each round, FEDFA incurs additional communication costs since it requires the sending 1) *from client to server* the momentum feature statistics $\bar{\mu}_m^k$ and $\bar{\sigma}_m^k$, as well as 2) *from server to client* the client sharing feature statistic variances $\gamma_{\mu^k}$ and $\gamma_{\sigma^k}$ at each layer $k$. Thus, for $K$ layers in total, the extra communication cost for each client is $c_e = 4 \sum_{k=1}^{K} C_k$, where the factor of $4$ is for server receiving and sending two statistic values. We further denote $c_m$ as the cost for exchanging model parameters in FedAvg. In general we have $c_e \ll c_m$ (*e.g.*, $c_e = 18$KB vs. $c_m = 99$MB for AlexNet), hence, the extra communication bruden in FEDFA is almost negligible.

## 5 RELATED WORK

**Federated Learning.** Recent years have witnessed tremendous progress in federated learning (Konečnỳ et al., 2016), which opens the door for privacy-preserving deep learning (Shokri & Shmatikov, 2015), *i.e.*, train a global model on distributed datasets without disclosing private data information. FEDAVG (McMahan et al., 2017) is a milestone; it trains local models independently in multiple clients and then averages the resulting model updates via a central server once in a while. However, FEDAVG is designed for i.i.d. data and suffers in statistical accuracy or even diverge if deployed over non-i.i.d. client samples. To address this issue, numerous efforts have been devoted to handling heterogeneous federated environments, by, for example, adding a dynamic regularizer to local objectives in FEDPROX (Li et al., 2020a) and FEDDYN (Acar et al., 2021), correcting client drift through variance reduction in SCAFFOLD (Karimireddy et al., 2020), adaptive server optimization in FEDOPT (Reddi et al., 2021), local batch normalization in FEDBN (Li et al., 2020b), or training a perturbed loss in FEDSAM (Qu et al., 2022) and FEDROBUST (Reisizadeh et al., 2020).

FEDMIX (Yoon et al., 2021), as far as we know, is the only existing method that solves federated learning based on data augmentation. It adapts the well-known MIXUP algorithm (Zhang et al., 2018) from centralized learning into the federated learning scenario. Nevertheless, FEDMIX requires exchanging local data (or averaged version) across clients for data interpolation, thereby suffering privacy issues. In addition, FEDMIX operates on the input level, while our approach focuses on latent feature statistic augmentation. Since deeper representations tend to disentangle the underlying factors of variation better (Bengio et al., 2013), traversing along latent space will potentially make our method encounter more realistic samples. This is supported by the fact that FEDFA achieves consistent performance improvements over FEDMIX in diverse scenarios.

**Data Augmentation.** Data augmentation has a long and rich history in machine learning. Early studies (Schölkopf et al., 1996; Kukačka et al., 2017) focus on *label-preserving* transformations to employ regularization via data, alleviating overfitting and improving generalization. For image data, some techniques, like random horizontal flipping and cropping are commonly used for training of advanced neural networks (He et al., 2016). In addition, there is a recent trend for *label-perturbing* augmentation, *e.g.*, MIXUP (Zhang et al., 2018) or CUTMIX (Yun et al., 2019). Separate from these input-level augmentation techniques are feature augmentation methods (Verma et al., 2019; Li et al., 2021a; 2022b; Zhou et al., 2021) that make augmentation in latent feature space. These various data augmentation techniques have shown great successes to learn domain-invariant models in the centralized setup. Our method is an instance of label-preserving feature augmentation, designed for federated learning. It is inspired by recent efforts on implicit feature augmentation (Li et al., 2021a; 2022b; Zhou et al., 2021) that synthesize samples of novel domains by manipulating instance-level feature statistics. In these works, feature statistics are treated as 'features', which capture essential domain-specific characteristics. In FEDFA, we estimate appropriate variances of feature statistics from a federated perspective, and draw novel statistics probablistically from a distribution centered on old statistics, while spanning with the variances. FEDFA avoids statistics mixing of instances from different clients, as done in FEDMIX (Yoon et al., 2021), thus can better preserve data privacy.

## 6 CONCLUSION

This work solves federated learning from a unique perspective of feature augmentation, yielding a new algorithm FEDFA that shows strong performance across various federated learning scenarios. FEDFA is based on a Gaussian modeling of feature statistic augmentation, where Gaussian variances are estimated in a federated manner, based on both local feature statistic distribution within each client, as well as universal feature statistic distribution across clients. We identify the implicit federation-aware regularization effects of FEDFA through theoretical analysis, and confirm its empirical superiority across a suite of benchmarks in federated learning.

## 7 REPRODUCIBILITY

Throughout the paper we have provided details facilitating reproduction of our empirical results. All our experiments are ran with a single GPU (we used NVIDIA GeForce RTX 2080 Ti with a 11G memory), thus can be reproduced by researchers with computational constraints as well. The source code has been made publicly available in https://github.com/tfzhou/FedFA. For the theoretical results, all assumptions, proofs and relevant discussions are provided in the Appendix.

## 8 ACKNOWLEDGMENTS

We would like to thank the anonymous referees for their valuable comments for improving the paper. This study was partly supported by Varian Research, Switzerland Grant.

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

This appendix provides theoretical proofs, additional results and experimental details for the main paper. It is organized in five sections:

- §A summarizes the algorithms of FEDFA in Algorithm 1 and FFA in Algorithm 2;
- §B presents detailed theoretical analysis and proofs of our approach;
- §C shows additional ablative experiments;
- §D provides a more detailed analysis of extra communication cost required by FEDFA;
- §E describes experimental details and more results.

## A    DETAILED ALGORITHM

In Algorithm 1, we illustrate the detailed training procedure of our FEDFA. It is consistent with algorithms such as FEDAVG (McMahan et al., 2017). In each communication round, the client performs local model training of the feature extractor $h(\boldsymbol{w}_h)$ and classifier $g(\boldsymbol{w}_g)$. We append a FFA layer (Algorithm 2) after each convolutional stage $h^k$. Each client additionally maintains a pair of momentum feature statistics $\{\bar{\mu}_m, \bar{\sigma}_m\}$, which is updated in a momentum manner during training. The parameters from local training (*i.e.*, $\boldsymbol{w} = \{\boldsymbol{w}_h, \boldsymbol{w}_g\}$), which are omitted in Algorithm 1, along with the momentum feature statistics are sent to server for model aggregation and computation of client-sharing statistic variances, which will be distributed back to clients for the next round of local training.

---

**Algorithm 1** FEDFA: federated training phase. (We omit the parameter updating procedure, which is exactly same to FEDAVG.)

---

**Input:** Number of clients $M$; number of communication rounds $T$; neural network $f = g \circ h$; each $\hat{\boldsymbol{X}}_m^0$ represents the collection of training images in corresponding clients;
**Output:** $\gamma_{\mu^k}, \gamma_{\sigma^k}$;
1: **for** $t = 1, 2, \ldots, T$ **do**
2:     **for each** client $m \in [M]$ **do**
3:         $\bar{\mu}_m = \mathbf{0}, \bar{\sigma}_m = \mathbf{1}$          ▷ Initialize averaged feature statistics for the client
4:         **for each** layer $k \in [K]$ **do**
5:             $\boldsymbol{X}_m^k = h^k(\hat{\boldsymbol{X}}_m^{k-1})$          ▷ Run the $k$-th layer of the feature extractor $h$
6:             $\hat{\boldsymbol{X}}_m^k, \bar{\mu}_m^k, \bar{\sigma}_m^k = \text{FFA}(\boldsymbol{X}_m^k, \bar{\mu}_m^k, \bar{\sigma}_m^k)$          ▷ FFA layer in Algorithm 2
7:         $Y = g(\hat{\boldsymbol{X}}_m^K)$          ▷ Run classifer $g$ to get predictions
8:         Run loss computation and backward optimization
9:     $\Sigma_{\mu^k}^2 = \frac{1}{M}\sum_{m=1}^M (\bar{\mu}_m^k - \mathbb{E}[\bar{\mu}^k])^2$          ▷ Compute client sharing statistic variance (Eq. 5)
10:     $\Sigma_{\sigma^k}^2 = \frac{1}{M}\sum_{m=1}^M (\bar{\sigma}_m^k - \mathbb{E}[\bar{\sigma}^k])^2$
11:     **for each** channel $j \in [C]$ **do**          ▷ Compute adaptive fusion coefficients (Eq. 6)
12:         $\gamma_{\mu^k}^{(j)} = \frac{C(1 + 1/\Sigma_{\mu^k}^{2,(j)})^{-1}}{\sum_{c=1}^C (1 + 1/\Sigma_{\mu^k}^{2,(c)})^{-1}}$
13:         $\gamma_{\sigma^k}^{(j)} = \frac{C(1 + 1/\Sigma_{\sigma^k}^{2,(j)})^{-1}}{\sum_{c=1}^C (1 + 1/\Sigma_{\sigma^k}^{2,(c)})^{-1}}$
14: **return** $\gamma_{\mu^k}, \gamma_{\sigma^k}$

---

## B    THEORETICAL INSIGHTS

In this section, we provide mathematical analysis to understand FEDFA. We begin with interpreting FEDFA as a noise injection process (Bishop, 1995; Camuto et al., 2020; Lim et al., 2022), which is a case of VRM (§2.1), and show that FEDFA injects federation-aware noises to latent representations (§B.1). Next, we demonstrate that, induced by federation-aware noise injection, FEDFA exhibits a natural form of federation-aware implicit regularization to local client training (§B.2). Without loss of generality, we conduct all analysis for an arbitrary client $m \in [M]$.

---

**Algorithm 2** Algorithm description of FFA for the $k$th layer in client $m$.

---

**Input:** Original feature $\boldsymbol{X}_m^k \in \mathbb{R}^{B \times C \times H \times W}$; momentum $\alpha = 0.99$; probability $p = 0.5$;
   Client-sharing fusion coefficients $\gamma_{\mu^k} \in \mathbb{R}^C$ and $\gamma_{\sigma^k} \in \mathbb{R}^C$ downloaded from the server;
   Accumulated feature statistics $\bar{\mu}_m^k \in \mathbb{R}^C$ and $\bar{\sigma}_m^k \in \mathbb{R}^C$;
**Output:** Augmented feature $\hat{\boldsymbol{X}}_m^k, \bar{\mu}_m^k, \bar{\sigma}_m^k$;
  1: **if** np.random.random() $< p$ **then**
  2: $\quad$ $\mu_m^k = \frac{1}{HW} \sum_{h=1}^H \sum_{w=1}^W \boldsymbol{X}_m^{k,(h,w)}$ $\qquad$ ▷ Compute channel-wise feature statistics (Eq. 2)
  3: $\quad$ $\sigma_m^k = \sqrt{\frac{1}{HW} \sum_{h=1}^H \sum_{w=1}^W (\boldsymbol{X}_m^{k,(h,w)} - \mu_m^k)^2}$
  4: $\quad$ $\Sigma_{\mu_m^k}^2 = \frac{1}{B} \sum_{b=1}^B (\mu_m^k - \mathbb{E}[\mu_m^k])^2$ $\qquad$ ▷ Compute client specific statistic variances (Eq. 3)
  5: $\quad$ $\Sigma_{\sigma_m^k}^2 = \frac{1}{B} \sum_{b=1}^B (\sigma_m^k - \mathbb{E}[\sigma_m^k])^2$
  6: $\quad$ $\hat{\Sigma}_{\mu_m^k}^2 = (\gamma_{\mu^k} + 1)\Sigma_{\mu_m^k}^2$ $\qquad$ ▷ Adaptive variance fusion (Eq. 7)
  7: $\quad$ $\hat{\Sigma}_{\sigma_m^k}^2 = (\gamma_{\sigma^k} + 1)\Sigma_{\sigma_m^k}^2$
  8: $\quad$ $\hat{\mu}_m^k = \mu_m^k + \epsilon_\mu \hat{\Sigma}_{\mu_m^k}$ $\qquad$ ▷ Sampling novel feature statistics (Eq. 9)
  9: $\quad$ $\hat{\sigma}_m^k = \sigma_m^k + \epsilon_\sigma \hat{\Sigma}_{\sigma_m^k}$
 10: $\quad$ $\hat{\boldsymbol{X}}_m^k = \hat{\sigma}_m^k \frac{\boldsymbol{X}_m^k - \mu_m^k}{\sigma_m^k} + \hat{\mu}_m^k$ $\qquad$ ▷ Transform original feature based on novel statistics (Eq. 8)
 11: $\quad$ $\bar{\mu}_m^k \leftarrow \alpha\bar{\mu}_m^k + (1-\alpha)\frac{1}{B}\sum_{b=1}^B \mu_m^k$ $\qquad$ ▷ Momentum updating feature statistics (Eq. 5)
 12: $\quad$ $\bar{\sigma}_m^k \leftarrow \alpha\bar{\sigma}_m^k + (1-\alpha)\frac{1}{B}\sum_{b=1}^B \sigma_m^k$
 13: **return** $\hat{\boldsymbol{X}}_m^k, \bar{\mu}_m^k, \bar{\sigma}_m^k$

---

### B.1 UNDERSTANDING FEDFA AS FEDERATION-AWARE NOISE INJECTION

**Noise Injection in Neural Networks.** Let $x$ be a training sample and $\boldsymbol{x}^k$ its latent representation at the $k$-th layer, with no noise injections. The $\boldsymbol{x}^k$ can be noised under a process $\hat{\boldsymbol{x}}^k = \boldsymbol{x}^k + \boldsymbol{e}^k$, where $\boldsymbol{e}^k$ is an addition noise drawn from a probability distribution, and $\boldsymbol{x}^k$ is the noised representation.

A popular choice of $\boldsymbol{e}^k$ is isotropic Gaussian noise (Camuto et al., 2020), *i.e.*, $\boldsymbol{e}^k \sim \mathcal{N}(0, \sigma^2 \boldsymbol{I})$, where $\boldsymbol{I}$ is an identity matrix and $\sigma$ is a scalar, controlling the amplitude of $\boldsymbol{e}^k$. To avoid over-perturbation that may cause model collapse, $\sigma$ is typically set as a small value. Despite its simplicity, the strategy is confirmed as a highly effective regularized for tackling domain generalization (Li et al., 2021b) and adversarial samples (Lecuyer et al., 2019; Cohen et al., 2019). However, as shown in Table 5, its performance (see FEDFA-R) is only marginally better or sometimes worse than FEDAVG in FL.

**Federation-Aware Noise Injection.** From Eq. 9, we can clearly see that the feature statistic augmentation in our approach follows the noise injection process above. Next we show that this eventually results in features perturbed under a federation-aware noising process.

**Lemma 1.** *Consider client $m$, for a batch-wise latent feature $\boldsymbol{X}_m^k$ at the $k$-th layer, its augmentation in* FEDFA *follows a noising process $\hat{\boldsymbol{X}}_m^k = \boldsymbol{X}_m^k + \boldsymbol{e}_m^k$, with the noise $\boldsymbol{e}_m^k$ taking the form:*

$$\boldsymbol{e}_m^k = \epsilon_\sigma \hat{\Sigma}_{\sigma_m^k} \bar{\boldsymbol{X}}_m^k + \epsilon_\mu \hat{\Sigma}_{\mu_m^k}, \tag{14}$$

*where $\epsilon_\mu \sim \mathcal{N}(0,1)$, $\epsilon_\sigma \sim \mathcal{N}(0,1)$, $\bar{\boldsymbol{X}}_m^k = (\boldsymbol{X}_m^k - \mu_m^k)/\sigma_m^k$.*

*Proof of Lemma 1.* We can easily prove this by substituting Eq. 9 into Eq. 8:

$$\hat{\boldsymbol{X}}_m^k = \hat{\sigma}_m^k \frac{\boldsymbol{X}_m^k - \mu_m^k}{\sigma_m^k} + \hat{\mu}_m^k,$$

$$= (\sigma_m^k + \epsilon_\sigma \hat{\Sigma}_{\sigma_m^k}) \frac{\boldsymbol{X}_m^k - \mu_m^k}{\sigma_m^k} + (\mu_m^k + \epsilon_\mu \hat{\Sigma}_{\mu_m^k}),$$

$$= \boldsymbol{X}_m^k + \underbrace{(\epsilon_\sigma \hat{\Sigma}_{\sigma_m^k} \bar{\boldsymbol{X}}_m^k + \epsilon_\mu \hat{\Sigma}_{\mu_m^k})}_{\boldsymbol{e}_m^k}, \tag{15}$$

$$\text{where} \quad \epsilon_\mu \sim \mathcal{N}(0,1), \quad \epsilon_\sigma \sim \mathcal{N}(0,1), \quad \bar{\boldsymbol{X}}_m^k = (\boldsymbol{X}_m^k - \mu_m^k)/\sigma_m^k.$$

As compared to the Gaussian noise injections (Camuto et al., 2020; Cohen et al., 2019; Lecuyer et al., 2019), the noise term $e_m^k$ in FEDFA shows several desirable properties: it is 1) data-dependent, adaptively determined based on the normalized input feature $\bar{X}_m^k$; 2) channel-independent, allowing for more extensive exploration along different directions in the feature space; 3) most importantly federation-aware, *i.e.*, its strength is controlled by statistic variances $\hat{\Sigma}_{\mu_m^k}$ and $\hat{\Sigma}_{\sigma_m^k}$ (*cf.* Eq. 7), which are known carrying universal statistic information of all participating clients.

## B.2 FEDERATION-AWARE IMPLICIT REGULARIZATION IN FEDFA

Next we show that with noise injections, FEDFA imposes *federation-aware implicit regularization* to local client training. By this, we mean regularization imposed implicitly by the stochastic learning strategy, without explicit modification of the loss, and the regularization effect is affected by the federation-aware noise (Lemma 1).

Recall the deep neural network $f$ defined in §2.1: $f \triangleq g \circ h$, where $h = h^K \circ h^{K-1} \circ \cdots \circ h^1$ is a $K$-layer CNN feature extractor and $g$ is a classifier. Given a batch of samples $X_m$ with labels $Y_m$, its latent representation at the $k$-th layer is computed as $X_m^k = h^k \circ h^{k-1} \circ \cdots \circ h^1(X_m)$, or we write it in a simpler form, $X_m^k = h^{1:k}(X_m)$. Note that we only add noises to layers in $h$, but not to $g$. Concretely, in each mini-batch training, FEDFA follows a stochastic optimization strategy to randomly select a subset of layers from $\{h^k\}_{k=1}^K$ and add noises to them. For simplicity, we denote $\mathcal{K} = \{1, \ldots, K\}$ as the index of all layers in $h$, $\mathcal{Z} \subseteq \mathcal{K}$ as the subset of layer indexes that are selected, $\mathcal{E} = \{e_m^z\}_{\forall z \in \mathcal{Z}}$ as the corresponding set of noises. Then, the loss function $\mathcal{L}_m^{\text{FEDFA}}$ of client $m$ in FEDFA can be equivalently written as $\mathcal{L}_m^{\text{FEDFA}} = \mathbb{E}_{\mathcal{Z} \sim \mathcal{K}} \mathcal{L}_m^{\mathcal{Z}}$, where $\mathcal{L}_m^{\mathcal{Z}}$ is a standard loss function $\mathcal{L}_m^{\text{ERM}}$ (*cf.* Eq. 1) imposed by adding noises to layers in $\mathcal{Z}$. In the remainder, we relate the loss function $\mathcal{L}_m^{\text{FEDFA}}$ to the original ERM loss $\mathcal{L}_m^{\text{ERM}}$ as well as a regularization term conditioned on $\mathcal{E}$.

**Theorem 1.** *In* FEDFA*, the loss function $\mathcal{L}_m^{\text{FEDFA}}$ of client $m$ can be expressed as:*

$$\mathcal{L}_m^{\text{FEDFA}} = \mathcal{L}_m^{\text{ERM}} + \mathcal{L}_m^{\text{REG}}, \tag{16}$$

*where $\mathcal{L}_m^{\text{ERM}}$ is the standard ERM loss, and $\mathcal{L}_m^{\text{REG}}$ is the regularization term:*

$$\mathcal{L}_m^{\text{ERM}} = \mathbb{E}_{(X_m, Y_m) \sim \mathbb{P}_m} \ell(g(h^{1:K}(X_m)), Y_m), \tag{17}$$

$$\mathcal{L}_m^{\text{REG}} = \mathbb{E}_{\mathcal{Z} \sim \mathcal{K}} \mathbb{E}_{(X_m, Y_m) \sim \mathbb{P}_m} \bigtriangledown_{h^{1:K}(X_m)} \ell(g(h^{1:K}(X_m)), Y_m)^\top \sum_{z \in \mathcal{Z}} \boldsymbol{J}^z(X_m) e_m^z, \tag{18}$$

*where $\boldsymbol{J}^z$ denotes the Jacobian of layer $z$ (defined in Proposition 1).*

Theorem 1 implies that, FEDFA implicitly introduces regularization to local client learning by regularizing the gradients of latent representations (*i.e.*, $\bigtriangledown_{h^{1:K}(X_m)} \ell(g(h^{1:K}(X_m)), Y_m)^\top$), weighted by federation-aware noises $\sum_{z \in \mathcal{Z}} \boldsymbol{J}^z(X_m) e_m^z$.

***In the remainder of this section, we prove Theorem 1.***

For the sake of analysis, we first marginalize the effect of the noising process. We do so by defining an *accumulated* noise $\hat{e}_m^K$ in the final layer $K$, which originates from the forward propagation of all noises in $\mathcal{E}$. We compute the accumulated noise based on (Camuto et al., 2020) that examines Gaussian noise injection into *every* latent layer in a neural network. Formally, the accumulated noise on the final convolutional layer $K$ can be expressed as follows:

**Proposition 1.** *Consider a $K$-layer neural network, in which a random noise $e_m^z$ is added to the activation of each layer $z \in \mathcal{Z}$. Assuming the Hessians, of the form $\bigtriangledown^2 h^{1:k}(X_m)|_{h^{1:n}(X_m)}$ where $k, n$ are the indexes over layers, are finite. Then, the accumulation noise $\hat{e}_m^K$ is approximated as:*

$$\hat{e}_m^K = \left( \sum_{z \in \mathcal{Z}} \boldsymbol{J}^z(X_m) e_m^z + O(\beta) \right), \tag{19}$$

*where $\boldsymbol{J}^z \in \mathbb{R}^{C_K \times C_z}$ indicates the Jacobian of layer $z$, i.e., $\boldsymbol{J}^z(X)_{i,j} = \frac{\partial h^{1:K}(X_m)_i}{\partial h^{1:z}(X_m)_j}$, where $C_K$ and $C_z$ denote the number of neurons in layer $K$ and $z$, respectively. $O(\beta)$ represents higher order terms in $\mathcal{E}$ that tend to be zero in the limit of small noises.*

*Proof of Proposition 1.* Starting with layer 1 as the first convolution layer, the accumulated noise on layer $K$ can be approximated through recursion. If $K = 1$, the accumulated noise is equal to $\hat{e}_m^K = e_m^1$. For $K = 2$, we apply Taylor's theorem on $h^2(\boldsymbol{X}_m^1 + e_m^1)$ around the output feature $\boldsymbol{X}_m^1$ at $h^1$. If we assume that all values in Hessian of $h^2(\boldsymbol{X}_m^1)$ is finite, the following approximation holds:

$$h^2(\boldsymbol{X}_m^1 + e_m^1) = h^2(\boldsymbol{X}_m^1) + \frac{\partial h^2(\boldsymbol{X}_m^1)}{\partial \boldsymbol{X}_m^1} e_m^1 + O(\kappa_1), \tag{20}$$

where $O(\kappa_1)$ denotes asymptotically dominated higher order terms given the small noise. In this special case of $K = 2$, we obtain the accumulated noise as

$$\hat{e}_m^K = \left( \frac{\partial h^2(\boldsymbol{X}_m^1)}{\partial \boldsymbol{X}_m^1} e_m^1 + O(\kappa_1) \right) + \check{e}_m^2. \tag{21}$$

The noise consists of two components: $\left( \frac{\partial h^2(\boldsymbol{X}_m^1)}{\partial \boldsymbol{X}_m^1} e_m^1 + O(\kappa_1) \right)$ is the noise propagated from $h^1$, while $\check{e}_m^2 = e_m^2$ is the noise added to $h^2$ if the layer is activated; otherwise, $\check{e}_m^2 = 0$. Note that Eq. 20 can be generalized to an arbitrary layer.

Repeating this process for each layer $z \in \mathcal{Z}$, and assuming that all Hessians of the form $\nabla^2 h^k(X_m)|_{h^n(X_m)}, \forall k < n$ are finite, we obtain the accumulated noise for layer $K$ as

$$\hat{e}_m^K = \left( \sum_{z \in \mathcal{Z} \backslash K} \frac{\partial h^{1:K}(X_m)}{\partial h^{1:z}(X_m)} e_m^z + O(\beta) \right) + \check{e}_m^K = \left( \sum_{z \in \mathcal{Z}} \frac{\partial h^{1:K}(X_m)}{\partial h^{1:z}(X_m)} e_m^z + O(\beta) \right), \tag{22}$$

where $\check{e}_m^K = e_m^K$ is the noise added to $h^K$ if layer $K$ is activated; otherwise, $\check{e}_m^K = 0$.

Denoting $\frac{\partial h^{1:K}(X_m)}{\partial h^{1:z}(X_m)}$ as the Jacobian $\boldsymbol{J}^z(X_m)$ completes the proof.

Based on Proposition 1, we provide a linear approximation of the loss $\ell$ for samples $(X_m, Y_m)$ as

$$\begin{aligned}
&\ell(g(h^{1:K}(X_m) + \hat{e}_m^K), Y_m) \\
&\approx \ell(g(h^{1:K}(X_m)), Y_m) + \nabla_{h^{1:K}(X_m)} \ell(g(h^{1:K}(X_m)), Y_m)^\top \hat{e}_m^K \\
&= \ell(g(h^{1:K}(X_m)), Y_m) + \nabla_{h^{1:K}(X_m)} \ell(g(h^{1:K}(X_m)), Y_m)^\top \sum_{z \in \mathcal{Z}} \boldsymbol{J}^z(X_m) e_m^z,
\end{aligned} \tag{23}$$

in which the higher order terms in Proposition 1 are neglected.

Based on Eq. 23, we further approximate the local training objective $\mathcal{L}_m^{\text{FEDFA}}$ in client $m$ and derive the regularization term as follows:

$$\begin{aligned}
\mathcal{L}_m^{\text{FEDFA}} &= \mathbb{E}_{\mathcal{Z} \sim \mathcal{K}} \mathcal{L}_m^{\mathcal{Z}} \\
&= \mathbb{E}_{\mathcal{Z} \sim \mathcal{K}} \mathbb{E}_{(X_m, Y_m) \sim \mathbb{P}_m} [\ell(g(h^{1:K}(X_m) + \hat{e}_m^K), Y_m)] \\
&= \mathbb{E}_{\mathcal{Z} \sim \mathcal{K}} \mathbb{E}_{(X_m, Y_m) \sim \mathbb{P}_m} [\ell(g(h^{1:K}(X_m)), Y_m) + \\
&\qquad\qquad \nabla_{h^{1:K}(X_m)} \ell(g(h^{1:K}(X_m)), Y_m)^\top \sum_{z \in \mathcal{Z}} \boldsymbol{J}^z(X_m) e_m^z] \\
&= \underbrace{\mathbb{E}_{(X_m, Y_m) \sim \mathbb{P}_m} \ell(g(h^{1:K}(X_m)), Y_m)}_{\mathcal{L}_m^{\text{ERM}}} + \\
&\quad \underbrace{\mathbb{E}_{\mathcal{Z} \sim \mathcal{K}} \mathbb{E}_{(X_m, Y_m) \sim \mathbb{P}_m} \nabla_{h^{1:K}(X_m)} \ell(g(h^{1:K}(X_m)), Y_m)^\top \sum_{z \in \mathcal{Z}} \boldsymbol{J}^z(X_m) e_m^z}_{\mathcal{L}_m^{\text{REG}}}.
\end{aligned} \tag{24}$$

Table 9: A summary of key experimental configuration for each dataset.

| Hyper-parameters | Office-Caltech 10 | DomainNet | ProstateMRI | EMNIST | CIFAR-10 |
|---|---|---|---|---|---|
| *federation-aware configuration* | | | | | |
| Number of rounds | 400 | 400 | 500 | 200 | 100 |
| Local training epochs | 1 | 1 | 1 | 10 | 10 |
| Number of clients | 4 | 6 | 6 | 100 | 100 |
| Participation rate | 1.0 | 1.0 | 1.0 | 0.1 | 0.1 |
| Number of total classes | 10 | 10 | 2 | 62 | 10 |
| *local client training configuration* | | | | | |
| Network | AlxeNet | AlexNet | U-Net | LeNet | ResNet |
| Optimizer | SGD | SGD | Adam | SGD | SGD |
| Local batch size | 32 | 32 | 16 | 64 | 10 |
| Local learning rate | 1e-2 | 1e-2 | 1e-4 | 1e-1 | 1e-1 |

## C ADDITIONAL ABLATION STUDY

In this section, we study the sensitivity of FedFA to the set of eligible layers to apply FFA. For notation, we use $\{1\}$ to represent that FFA is applied to the 1st convolutional stage; $\{1, 2\}$ to represent that FFA is applied to both the 1st and 2nd convolutional stages; and so forth. The results are shown in Table 8. We observe that i) our default design (using five layers) always shows the best performance on the three datasets (Office, DomainNet and ProstateMRI). We conjecture that this is due to its potential to beget more comprehensive augmentation; ii) applying FFA to only one particular layer brings minor gains against FedAvg; but iii) by adding more layers, the performance tends to improve. This implies that our approach benefits from inherent complementarity of features in different network layers.

Table 8: Performance for different sets of eligible layers to apply FFA.

| Variant | Office | DomainNet | ProstateMRI |
|---|---|---|---|
| FEDAVG | 78.5 | 62.8 | 86.5 |
| $\{1\}$ | 78.8 | 63.5 | 88.5 |
| $\{1, 2\}$ | 80.0 | 63.9 | 88.6 |
| $\{1, 2, 3\}$ | 80.0 | 64.0 | 89.0 |
| $\{1, 2, 3, 4\}$ | 80.6 | 64.3 | 88.8 |
| $\{1, 2, 3, 4, 5\}$ | **83.1** | **66.5** | **89.5** |
| $\{2, 3, 4, 5\}$ | 81.6 | 65.2 | 88.6 |
| $\{1, 2, 4, 5\}$ | 82.0 | 65.8 | 88.8 |
| $\{3, 4, 5\}$ | 78.4 | 63.8 | 87.0 |
| $\{4, 5\}$ | 79.4 | 64.7 | 85.9 |
| $\{5\}$ | 79.2 | 64.6 | 86.3 |
| $\{1, 5\}$ | 80.4 | 65.5 | 88.5 |
| $\{2, 3, 4\}$ | 79.5 | 64.3 | 88.8 |
| $\{2, 3\}$ | 78.7 | 64.0 | 88.5 |
| $\{3, 4\}$ | 78.3 | 63.2 | 86.5 |
| $\{3\}$ | 78.0 | 63.1 | 86.5 |

## D ANALYSIS OF ADDITIONAL COMMUNICATION COST IN FEDFA

In each round, FEDFA incurs additional communication costs since it requires the sending 1) *from client to server* the momentum feature statistics $\bar{\mu}_m^k$ and $\bar{\sigma}_m^k$, as well as 2) *from server to client* the client sharing feature statistic variances $\gamma_{\mu^k}$ and $\gamma_{\sigma^k}$ at each layer $k$. Thus, for $K$ layers in total, the extra communication cost for each client is $c_e = 4 \sum_{k=1}^{K} C_k$, where the factor of 4 is for server receiving and sending two statistic values. As presented in Table 10 and Table 14, we append one FFA layer after each convolutional stage of feature extractors in AlexNet and U-Net. Hence, the total additional communication costs for AlexNet and U-Net are:

$$\text{AlexNet: } 4 \times (64 + 192 + 384 + 256 + 256)/1024 \times 4 = 18 \text{ KB},$$
$$\text{U-Net: } 4 \times (32 + 64 + 128 + 256 + 512)/1024 \times 4 = 15.5 \text{ KB}. \tag{25}$$

However, it should be noted that these additional costs are minor in comparison with the cost required for exchanging model parameters, which are $2 \times 49.5$ MB and $2 \times 29.6$ MB for AlexNet and U-Net, respectively.

# E  EXPERIMENTAL DETAILS

## E.1  DATASET

We conduct extensive experiments on five datasets:

**Office-Caltech 10** (Gong et al., 2012) has four data sources, three from Office-31 (Saenko et al., 2010) and one from Caltech-256 (Griffin et al., 2007). They are collected from different camera devices or in diverse environments with different background.

**DomainNet** (Peng et al., 2019) contains images from six domains (clipart, infograph, painting, real, and sketch), which are collected by searching a category name along with a domain name in different search engines.

**ProstateMRI** (Liu et al., 2020b) is a multi-site prostate segmentation dataset consisting of six data sources of T2-weighted MRI from different medical institutions. For all the three datasets, we regard each data source as a client, and thus real-world feature shift exists among clients.

**CIFAR-10** (Krizhevsky & Hinton, 2009) is a popular natural image classification dataset for federated learning. It contains 50,000 training and 10,000 test images. We introduce label distribution heterogeneity for the dataset, by sampling local data based on the Dirichlet distribution $\text{Dir}(\alpha)$, and consider two different concentration parameters, *i.e.*, $\alpha = 0.6$ or $0.3$.

**EMNIST** (Cohen et al., 2017) is an image classification dataset with 62 classes, including all 26 capital and small letter of alphabet as well as numbers. We follow the setup in FEDMIX to simulate data size heterogeneity.

In Table 9, we summarize the configuration of our experiments for each of the datasets.

## E.2  EXPERIMENTAL DETAILS FOR IMAGE CLASSIFICATION

**Network Architecture.** For the image classification tasks on Office-Caltech 10 (Gong et al., 2012) and DomainNet (Peng et al., 2019), we use an adapted AlexNet (Krizhevsky et al., 2017), with the detailed network architecture shown in Table 10.

**Training Details.** For each training image in Office-Caltech10 and DomainNet, we reshape its size into $256 \times 256$. We train AlexNet with the SGD optimizier with a learning rate of $0.01$, a mini-batch size of 32, using the standard cross-entropy loss. The total number of communication round is set to 400, with one local epoch per round by default. Two basic data augmentation techniques are also applied for training, *i.e.*, random horizontal flipping and random rotation with degree in $[-30, 30]$. The dataset splits of Office-Caltech 10 and DomainNet in our experiments are summarized in Table 11 and Table 12, respectively.

Table 10: **Network architecture of AlexNet** for Office-Caltech10 and DomainNet experiments. For convolutional layer (Conv2D), we list parameters with sequence of input and output dimension, kernal size, stride and padding. For max pooling layer (MaxPool2D), we list kernal and stride. For fully connected layer (FC), we list input and output dimension. For Batch Normalization layer (BN), we list the channel dimension. Note that **FFA** denotes the proposed feature augmentation layer, and we list the dimension of its input feature.

| Layer | Details |
|-------|---------|
| 1 | Conv2D(3, 64, 11, 4, 2), BN(64), ReLU, MaxPool2D(3, 2) |
| 2 | **FFA**(64) |
| 3 | Conv2D(64, 192, 5, 1, 2), BN(192), ReLU, MaxPool2D(3, 2) |
| 4 | **FFA**(192) |
| 5 | Conv2D(192, 384, 3, 1, 1), BN(384), ReLU |
| 6 | **FFA**(384) |
| 7 | Conv2D(384, 256, 3, 1, 1), BN(256), ReLU |
| 8 | **FFA**(256) |
| 9 | Conv2D(256, 256, 3, 1, 1), BN(256), ReLU, MaxPool2D(3, 2) |
| 10 | **FFA**(256) |
| 11 | AdaptiveAvgPool2D(6, 6) |
| 12 | FC(9216, 1024), BN(1024), ReLU |
| 13 | FC(1024, 1024), BN(1024), ReLU |
| 14 | FC(1024, num_class) |

Table 11: Numbers of samples in the training, validation, and testing sets of each client in Office-Caltech 10 used in our experiments.

| Split | Amazon | Caltech | DSLR | Webcam |
|-------|--------|---------|------|--------|
| train | 459 | 538 | 75 | 141 |
| val | 307 | 360 | 50 | 95 |
| test | 192 | 225 | 32 | 59 |

Table 12: Numbers of samples in the training, validation, and testing sets of each client in DomainNet used in our experiments.

| Split | Clipart | Infograph | Painting | Quickdraw | Real | Sketch |
|-------|---------|-----------|----------|-----------|------|--------|
| train | 672 | 840 | 791 | 1280 | 1556 | 708 |
| val | 420 | 525 | 494 | 800 | 972 | 442 |
| test | 526 | 657 | 619 | 1000 | 1217 | 554 |

Table 13: Numbers of samples in the training, validation, and testing sets of each client in ProstateMRI used in our experiments.

| Split | BIDMC | HK | I2CVB | BMC | RUNMC | UCL |
|-------|-------|-----|-------|-----|-------|-----|
| train | 156 | 94 | 280 | 230 | 246 | 105 |
| val | 52 | 31 | 93 | 76 | 82 | 35 |
| test | 52 | 31 | 93 | 76 | 82 | 35 |

### E.3  EXPERIMENTAL DETAILS FOR MEDICAL IMAGE SEGMENTATION

**Network Architecture.** For medical image segmentation on ProstateMRI (Liu et al., 2020b), we use a vanilla U-Net architecture, as presented in Table 14 and Table 15.

**Training Details.** Following FEDHARMO, we use a combination of standard cross-entropy and Dice loss to train the network, using the Adam optimizer with learning rate 1e-4, batch size 16, and weight decay 1e-4. No any data augmentation techniques are applied. The dataset splits of ProstateMRI used in our experiments are summarized in Table 13

**Additional Results.** Table 16 provides a detailed performance statistic of different methods on ProstateMRI, w.r.t. different fractions ($1/6$, $2/6$, $3/6$, $4/6$, $5/6$, 1) of training samples used in each client. The table corresponds to the plot in Fig. 2.

Table 14: **Network architecture of U-Net** for medical image segmentation experiments on ProstateMRI. The structure of 'Block' module is provided in Table 15. Note that **FFA** denotes the proposed feature augmentation layer, and we list the dimension of its input feature.

| Layer | Details |
|-------|---------|
| 1 | Block(in_features=3, features=32, name="encoder1"), MaxPool2D(2, 2) |
| 2 | **FFA**(32) |
| 3 | Block(in_features=32, features=64, name="encoder2"), MaxPool2D(2, 2) |
| 4 | **FFA**(64) |
| 5 | Block(in_features=64, features=128, name="encoder3"), MaxPool2D(2, 2) |
| 6 | **FFA**(128) |
| 7 | Block(in_features=128, features=256, name="encoder4"), MaxPool2D(2, 2) |
| 8 | **FFA**(256) |
| 9 | Block(in_features=256, features=512, name="bottleneck") |
| 10 | **FFA**(512) |
| 11 | Block(in_features=512, features=256, name="decoder4") |
| 12 | Block(in_features=256, features=128, name="decoder3") |
| 13 | Block(in_features=128, features=64, name="decoder2") |
| 14 | Block(in_features=64, features=32, name="decoder1") |
| 15 | Conv2d(32, num_class, 1, 1) |

Table 15: **Detailed structure** of the 'Block' module in U-Net (Table 14)

| Layer | Details |
|-------|---------|
| 1 | Conv2d(in_features, features, 3, 1) |
| 2 | BatchNorm2d(features) |
| 3 | ReLU |
| 4 | Conv2d(features, features, 3, 1) |
| 5 | BatchNorm2d(features) |
| 6 | ReLU |

Table 16: **Segmentation performance on ProstateMRI `test`** (Liu et al., 2020b) in terms of Dice score (%). We sample different fractions ($[^1/_6, ^2/_6, ^3/_6, ^4/_6, ^5/_6, 1]$) of training samples over the original training set in each client to study the effects of methods w.r.t the variations of local training size. *The table provides a detailed statistic for Fig. 2.*

| Algorithm | BIDMC | HK | I2CVB | BMC | RUNMC | UCL | Average |
|---|---|---|---|---|---|---|---|
| *fraction of training samples over the whole training set: 1/6* | | | | | | | |
| FEDAVG (McMahan et al., 2017) | 81.2 | 90.8 | 86.1 | 84.0 | 91.0 | 86.2 | 86.5 |
| FEDPROX (Li et al., 2020a) | 82.8 | 89.1 | 89.8 | 79.4 | 89.8 | 85.6 | 86.1 |
| FEDAVGM (Hsu et al., 2019) | 80.3 | 91.6 | 88.2 | 82.2 | 91.2 | 86.5 | 86.7 |
| FEDSAM (Qu et al., 2022) | 82.7 | 92.5 | 91.8 | 83.6 | 92.6 | 88.1 | 88.5 |
| FEDHARMO (Jiang et al., 2022) | 86.7 | 91.6 | 92.7 | 84.2 | 92.5 | 84.6 | 88.7 |
| FEDMIX (Yoon et al., 2021) | 86.3 | 91.6 | 89.6 | 88.1 | 89.8 | 85.2 | 88.4 |
| FEDFA-R | 78.9 | 91.3 | 87.0 | 81.9 | 90.6 | 86.6 | 86.1 |
| FEDFA-C | 80.5 | 92.4 | 89.2 | 83.7 | 92.1 | 89.4 | 87.9 |
| **FEDFA** | 85.7 | 92.6 | 91.0 | 85.4 | 92.9 | 89.2 | **89.5** |
| *fraction of training samples over the whole training set: 2/6* | | | | | | | |
| FEDAVG (McMahan et al., 2017) | 83.5 | 92.4 | 90.1 | 86.5 | 93.7 | 89.4 | 89.3 |
| FEDPROX (Li et al., 2020a) | 83.9 | 92.7 | 93.7 | 86.2 | 94.0 | 88.6 | 89.8 |
| FEDAVGM (Hsu et al., 2019) | 80.8 | 91.8 | 91.7 | 83.5 | 93.5 | 86.1 | 87.9 |
| FEDSAM (Qu et al., 2022) | 83.6 | 93.7 | 94.3 | 86.6 | 94.7 | 88.2 | 90.2 |
| FEDHARMO (Jiang et al., 2022) | 85.6 | 90.8 | 92.5 | 88.5 | 94.3 | 87.6 | 89.9 |
| FEDMIX (Yoon et al., 2021) | 88.6 | 92.7 | 93.3 | 90.0 | 91.7 | 87.3 | 90.6 |
| FEDFA-R | 81.5 | 92.1 | 90.9 | 84.2 | 93.1 | 89.4 | 88.5 |
| FEDFA-C | 81.1 | 92.0 | 92.9 | 85.2 | 94.0 | 86.9 | 88.7 |
| **FEDFA** | 87.8 | 92.3 | 92.9 | 86.6 | 95.1 | 89.8 | **90.8** |
| *fraction of training samples over the whole training set: 3/6* | | | | | | | |
| FEDAVG (McMahan et al., 2017) | 84.2 | 90.9 | 93.9 | 87.0 | 93.8 | 89.1 | 89.8 |
| FEDPROX (Li et al., 2020a) | 82.0 | 92.5 | 92.0 | 84.7 | 92.3 | 88.1 | 88.6 |
| FEDAVGM (Hsu et al., 2019) | 81.8 | 92.0 | 94.4 | 85.7 | 93.5 | 88.7 | 89.3 |
| FEDSAM (Qu et al., 2022) | 85.1 | 92.9 | 95.0 | 88.2 | 95.4 | 90.3 | 91.2 |
| FEDHARMO (Jiang et al., 2022) | 92.2 | 90.9 | 96.3 | 89.8 | 95.0 | 86.7 | 91.8 |
| FEDMIX (Yoon et al., 2021) | 87.2 | 92.8 | 94.9 | 90.1 | 92.5 | 90.0 | 91.3 |
| FEDFA-R | 82.7 | 92.8 | 94.4 | 87.0 | 94.7 | 88.2 | 90.0 |
| FEDFA-C | 87.0 | 91.1 | 93.5 | 90.3 | 93.9 | 89.4 | 90.9 |
| **FEDFA** | 87.4 | 93.5 | 95.6 | 90.6 | 95.6 | 91.4 | **92.4** |
| *fraction of training samples over the whole training set: 4/6* | | | | | | | |
| FEDAVG (McMahan et al., 2017) | 86.4 | 92.3 | 95.7 | 87.7 | 95.2 | 89.3 | 91.1 |
| FEDPROX (Li et al., 2020a) | 87.2 | 92.9 | 94.0 | 85.8 | 94.5 | 89.7 | 90.7 |
| FEDAVGM (Hsu et al., 2019) | 85.6 | 93.0 | 95.0 | 84.9 | 94.8 | 87.3 | 90.1 |
| FEDSAM (Qu et al., 2022) | 91.3 | 93.8 | 95.8 | 91.1 | 96.0 | 92.5 | **93.4** |
| FEDHARMO (Jiang et al., 2022) | 91.0 | 94.2 | 94.3 | 90.3 | 95.8 | 92.2 | 93.0 |
| FEDMIX (Yoon et al., 2021) | 90.0 | 94.2 | 95.5 | 91.7 | 93.2 | 91.5 | 92.7 |
| FEDFA-R | 84.9 | 91.1 | 95.2 | 86.0 | 95.4 | 91.2 | 90.6 |
| FEDFA-C | 87.4 | 93.6 | 95.2 | 88.8 | 95.4 | 91.4 | 92.0 |
| **FEDFA** | 91.1 | 93.6 | 95.1 | 89.5 | 95.3 | 91.2 | 92.6 |
| *fraction of training samples over the whole training set: 5/6* | | | | | | | |
| FEDAVG (McMahan et al., 2017) | 89.2 | 94.2 | 94.8 | 89.0 | 95.1 | 91.3 | 92.3 |
| FEDPROX (Li et al., 2020a) | 88.7 | 94.7 | 94.8 | 88.9 | 95.6 | 91.1 | 92.3 |
| FEDAVGM (Hsu et al., 2019) | 87.7 | 94.6 | 94.3 | 89.3 | 95.2 | 90.4 | 91.9 |
| FEDSAM (Qu et al., 2022) | 91.9 | 95.2 | 96.1 | 90.7 | 96.3 | 91.2 | 93.5 |
| FEDHARMO (Jiang et al., 2022) | 89.8 | 94.5 | 95.6 | 91.0 | 96.0 | 92.1 | 93.2 |
| FEDMIX (Yoon et al., 2021) | 91.6 | 92.9 | 94.4 | 92.8 | 94.0 | 92.0 | 93.0 |
| FEDFA-R | 89.9 | 94.6 | 95.6 | 88.8 | 95.7 | 91.4 | 92.7 |
| FEDFA-C | 90.3 | 94.7 | 95.5 | 88.7 | 95.8 | 91.5 | 92.7 |
| **FEDFA** | 93.2 | 95.1 | 96.4 | 91.2 | 96.3 | 91.3 | **93.9** |
| *fraction of training samples over the whole training set: 1* | | | | | | | |
| FEDAVG (McMahan et al., 2017) | 89.5 | 94.0 | 95.6 | 87.5 | 95.1 | 91.1 | 92.1 |
| FEDPROX (Li et al., 2020a) | 88.2 | 94.1 | 95.6 | 88.8 | 95.9 | 91.6 | 92.4 |
| FEDAVGM (Hsu et al., 2019) | 92.1 | 94.2 | 96.0 | 87.4 | 95.0 | 90.6 | 92.6 |
| FEDSAM (Qu et al., 2022) | 92.0 | 95.4 | 96.6 | 90.1 | 96.3 | 92.2 | 93.8 |
| FEDHARMO (Jiang et al., 2022) | 87.4 | 95.0 | 95.5 | 91.2 | 96.1 | 92.2 | 92.9 |
| FEDMIX (Yoon et al., 2021) | 93.7 | 95.2 | 95.7 | 91.9 | 94.8 | 92.8 | 94.0 |
| FEDFA-R | 87.1 | 92.5 | 95.0 | 86.9 | 95.7 | 90.8 | 91.3 |
| FEDFA-C | 91.5 | 93.4 | 96.5 | 88.9 | 95.4 | 91.6 | 92.9 |
| **FEDFA** | 93.7 | 95.4 | 96.0 | 92.2 | 96.2 | 92.5 | **94.3** |

