# OpenReview forum: "FedFA:  Federated Feature Augmentation"
_ICLR.cc/2023/Conference — ICLR 2023 poster_

### Official Review · Reviewer_q8Hi · 2022-10-18

**Confidence:** 5
**Correctness:** 4
**Technical Novelty And Significance:** 4
**Empirical Novelty And Significance:** 4
**Recommendation:** 8

**Clarity, Quality, Novelty And Reproducibility:**

The paper clearly explains the concepts and the presentation is easy to follow

The work has quality but it can be further improved by increasing the scale of the benchmarks.

The proposed approach is simple and novel in the realm of federated optimization algorithms which definitely contributes to the body of knowledge in the domain/literature.

The method is easy to reproduce and the results can also be tested, having access to the open sourced code can help further.

**Strength And Weaknesses:**

# Pros:

- The paper is well-written, easy to understand and follow along.

- The idea of treating multiple channels as multiple variates and measuring the statistical variances is really interesting since they will always remain normally distributed as they are measured over mean and variances already.

- A very simple technique with no or negligible communication overheads.

- Literature is covered adequately.

# Cons:

- The FL based hyper params are missing in the paper, such as the number of rounds, the number of clients participating in training, client/device selection if any, etc.

- The scale of the benchmarks is small compared to CIFAR-10 or EMNIST, can you show the method works on one/both of these relatively large dataset.

- Also, the number of samples per client seems to be uniform and the proposed statistical measures seem to work well for something like that. Can you experiment with an unequal number of samples per client?

- The main advantage that I see with FEDFA are the communication overheads compared to mixup kind of augmentation approaches (of course privacy is there, this on top), why not asymptotically quantify the costs?The cost analysis that is currently presented is talking about some raw numbers, they will go away if it is \BigO notation and the costs might be almost same as FedAvg.

- It looks like there is an ablation on \alpha and \p done separately, but what is the best combination, can we call it (\alpha, p) = (1, 0.5) or (1.,0, 1.0) or (0.5,0.5)?


### Minor comments:

- “Since data coming different users … ” should be corrected


**Summary Of The Paper:**

Well known problem by now is the non-iidness of the client data causes degradations in FL trained models. The paper proposes FedFA, federated feature augmentation to address feature shift in clients data. FedFA exploits instance based statistics to augment local features to curb the introduction of novel domains because of augmentations. The proposed approach sounds promising to advance FL.

**Summary Of The Review:**

Please refer to the strengths and weaknesses

---

> ### Author Response · Authors · 2022-11-16
> **Response to Reviewer q8Hi (1/2)**
>
> We thank the reviewer for the feedback and suggestions,
> and appreciate the positive comments.
> We address your comments below, and start with the following question:
>
> > The scale of the benchmarks is small compared to CIFAR-10 or EMNIST,
> > can you show the method works on one/both of these relatively large dataset?
>
> Sure. New results on the two datasets are presented in Table 3 in the latest revision
> and copied as below for convenience. We briefly summarize the experimental setup and results here.
>
> We simulate two non-i.i.d environments:
> * For CIFAR-10, we sample local data using Dirichlet distribution Dir$(\alpha)$ by setting $\alpha$ to 0.6 or 0.3 to simulate different non-i.i.d. levels. This leads to __label distribution heterogeneity__.
> * For EMNIST, we follow FedMix to split data based on writers. This results in clients with varied numbers of local samples, yielding __data size heterogeneity__.
>
> As seen, FedFA shows leading performance in all cases.
> This suggests that FedFA is scalable to large-size datasets,
> and generalizable to  various non-i.i.d. challenges in FL.
>
> |Algorithm | CIFAR-10, Dir(0.6) | CIFAR-10, Dir(0.3) | EMNIST|
> |:--------|:------------------:|:------------------:|:-----:|
> |FedAvg    |73.3 | 69.2 | 84.9|
> |FedAvgM   |73.4 | 69.1 | 85.5|
> |FedProx   |74.0 | 69.5 | 84.9|
> |FedBN     |73.7 | 69.8 | 85.3|
> |FedSAM    |74.3 | 70.0 | 86.5|
> |FedRobust |74.9 | 70.5 | 86.7|
> |FedMix    |75.5 | 70.7 | 86.6|
> |FedFA     |**76.3** |**71.9** |**87.8** |
>
> Your follow-up question about evaluation is:
> > Also, the number of samples per client seems to be uniform and
> > the proposed statistical measures seem to work well for something like that.
> > Can you experiment with an unequal number of samples per client?
>
> This unbalanced situation has been covered by our experiments:
>
> One case is Office-Caltech 10, in which  sample sizes across the four clients vary a lot, i.e., 459, 538, 75, 141 (see Table 12).
> The results below (copied from Table 1) demonstrate the superiority of FedFA in dealing with such a unbalanced case.
> Notably, it outperforms the counterparts tremendously (__+8%__) in the  client (i.e., DSLR) with the smallest number of training samples.
>
> |Algorithm | Amazon (459) | Caltech (538) | DSLR (75) | Webcam (141) | Average |
> |:----------|:------------:|:------------:|:---------:|:------------:|:-------:|
> |FedAvg | 84.4 | 66.7 | 75.0 | 88.1 | 78.5 |
> |FedBN  | 82.3 | 63.6 | 81.2 | 94.9 | 80.5 |
> |FedMix | 81.7 | 63.1 | 81.3 | 93.2 | 79.8 |
> |FedFA  | __88.0__ | __65.8__ | __90.6__ | __88.1__ | __83.1__ |
>
> The conclusion is underpinned by our new experiments on EMNIST,
> for which we follow FedMix to purposely simulate a data size heterogeneity environment,
> as mentioned in the response to your previous comment.
>
> [Our response continues ...]

---

> > ### Author Response · Authors · 2022-11-16
> > **Response to Reviewer q8Hi (2/2)**
> >
> > In the following, we address all other remarks and suggestions:
> > > The FL based hyper params are missing in the paper, such as the number of rounds,
> > > the number of clients participating in training, client/device selection if any, etc.
> >
> > Thanks for pointing this out; we have provided all missing hyper-params in Section 4.1
> > and Table 10 in the appendix.
> >
> >
> > > The main advantage that I see with FedFA are the communication overheads
> > > compared to mixup kind of augmentation approaches (of course privacy is there, this on top),
> > > why not asymptotically quantify the costs?
> > > The cost analysis that is currently presented is talking about some raw numbers,
> > > they will go away if it is Big-O notation and the costs might be almost same as FedAvg.
> >
> > Thanks for the suggestion, and we agree that a more formal analysis would be better here.
> >
> > Let's denote the cost for exchange of model parameters as $cost_\text{model}$.
> > In FedFA, for each client, the extra communication cost in a round is $cost_\text{extra} = 4\sum_{k=1}^KC_k$.
> > Here, $K$ is the number of FFA layers, which is small and uniform for typical network architectures
> > (5 in all experiments); $C_k$ is the number of feature channel in layer $k$; the factor of $4$ is
> > for server receiving and sending two statistic values. Note that in general we have $cost_\text{extra} \ll cost_\text{model}$
> > (e.g., $cost_\text{extra}=\text{18KB}$ vs. $cost_\text{model}=\text{99MB}$ for AlexNet),
> > thus, the extra communication burden is almost negligible. For asymptotic analysis,
> > the extra cost is roughly $\mathcal{O}(C)$, while the number of model parameters,
> > taking AlexNet as an example, is approximately $\mathcal{O}(C^2)$, where $C=\text{max} {C_1,C_2,\ldots,C_K\}$.
> > This implies that the communication cost of FedFA is asymptotically same to FedAvg. Section 4.5 has been revised to reflect this analysis.
> >
> > > It looks like there is an ablation on $\alpha$ and $p$ done separately,
> > > but what is the best combination, can we call it $(\alpha, p)$ = (1, 0.5) or (1.,0, 1.0) or (0.5,0.5)?
> >
> > The two hyper-parameters are in nature independent of each other. Empirically, we also observe very weak correlations between them.
> > We are certain that the current ablation is sufficient to unveil their true effects. Thanks.
> >
> > > "Since data coming different users …" should be corrected
> >
> > Fixed. Thanks!

---

### Official Review · Reviewer_7jRY · 2022-10-21

**Confidence:** 4
**Correctness:** 3
**Technical Novelty And Significance:** 3
**Empirical Novelty And Significance:** 3
**Recommendation:** 6

**Clarity, Quality, Novelty And Reproducibility:**

This work is of fair quality. I feel the writing is not clear enough and its technical details need to be clarified. The originality of the work is good.

**Strength And Weaknesses:**

Strength:
-- modeling the feature statistic via a Gaussian distribution and generating new features by sampling from the Gaussian distribution seems interesting.
-- the client sharing statistic variances and adaptive variance fusion strategy are technically solid.

Weakness:
What concerns me is the writing of this work. I feel it is not clear enough and I feel a little hard to fully understand the technical details.
-- I feel difficult to fully understand the technical details. For example, in Eq 6, there is j  while in Eq 7 there isn't. How to use the result calculated in Eq 6 for Eq 7?
-- How to take advantage of the shared feature statistics are not clear to me. Specifically,  the relation between Eq. 5/6/7 is not clear to me.
-- the formulation terms in this draft need refined. For example, in page 4, \mu_m^k \sim \mathcal{N} (\mu_m^k, ##).

**Summary Of The Paper:**

This paper tries to address feature shift by modelling feature statistics among clients.  Specifically, the authors model feature statistic via a Gaussian distribution.  The mean of the Gaussian distribution denote the original statistic while the variance denote the augmentation scope. The authors designed a solution to determine an appropriate variance for the Gaussian distribution. In the solution, the Gaussian variance is estimated by not only the local data but also the feature statistics from all participating clients.  In the variance estimation, the authors designed an adaptive variance fusion strategy to assign different weights to different channels in the feature statistic space.
After establishing the Gaussian distribution, the novel features can be synthesised by sampling from the Gaussian distribution.

**Summary Of The Review:**

This work propose to model feature statistic via a Gaussian distribution. The parameters of the Gaussian distribution is estimated from the data from active clients. For the parameter estimation, the authors designed client sharing statistic variances and adaptive variance fusion strategies.  Once the parameters are estimated, new features are generated by sampling from the Gaussian distribution. Although the idea seems interesting, the writing of this work needs significant improvement. I feel hard to fully understand the technical details.

---

> ### Author Response · Authors · 2022-11-16
> **Response to Reviewer 7jRY**
>
> We appreciate the reviewer for pointing out certain issues of clarity in the technical part
> which we have revised in the latest revision and provide clarifications below.
>
> To begin with, we explain the relations between symbols $\gamma_{\mu^k}^{(j)}/\gamma_{\sigma^k}^{(j)}$ in Eq. 6, and $\gamma_{\mu_k}/\gamma_{\sigma_k}$ in Eq. 7, to answer your question:
> > In Eq. 6, there is $j$ while in Eq. 7 there isn't. How to use the results calculated in Eq. 6 for Eq. 7?
>
> Here $j$ is a channel index.
> That said, $\gamma_{\mu^k}^{(j)}\in\mathbb{R}$ ($\gamma_{\sigma^k}^{(j)}\in\mathbb{R}$)
> is the value of the $j$-th channel in $\gamma_{\mu_k}\in\mathbb{R}^C$ ($\gamma_{\sigma_k}\in\mathbb{R}^C$),
> or in a more formal way: $\gamma_{\mu^k}=[\gamma_{\mu^k}^{{(1)}},...,\gamma_{\mu^k}^{(C)}]$
> and $\gamma_{\sigma^k}=\[\gamma_{\sigma^k}^{(1)},...,\gamma_{\sigma^k}^{(C)}\]$.
> In the original writing, we accidentally omitted the explicit definition of $\gamma_{\mu_k}/\gamma_{\sigma_k}$,
> which has been fixed with some additional explanations (see page 5). Thanks.
>
> Furthermore, you mentioned:
> > How to take advantage of the shared feature statistics are not clear to me.
> > Specifically, the relation between Eqs. 5/6/7 is not clear to me.
>
> **Relations between Eqs.5/6/7:** both Eq. 5 and Eq. 6
> are for computing the shared feature statistic variances, and executed in the ___server___ side;
> their relation is: Eq. 6 modulates the output of Eq. 5 by the Student's t-distribution to further
> highlight channels with larger variances, which would drive more significant augmentation
> along the channels. In contrast, Eq. 7 works in the ___client___ side:
> it fuses the shared statistics from Eq. 6 as well as client specific statistics from Eq. 3 to yield plausible local statistic variances.
>
> We  agree that the writing of mentioned parts can be improved
> upon rereading. We have reorganized relevant paragraphs to hopefully
> make things much clearer. A primary action is to present Eq. 6 more
> immediately after Eq. 5, avoiding its entanglement with Eq. 7 (see page 5).
>
>
> > The formulation terms in this draft need refined.
> > For example, in page 4, $\mu_m^k\sim\mathcal{N}(\mu_m^k, \hat{\Sigma}_{\mu_m^k}^2)$ and
> >
> > $\sigma_m^k \sim \mathcal{N} (\sigma_m^k, \hat{\Sigma}^2_{\sigma_m^k})$.
>
> Thanks for your careful review. The formulations have been corrected as $\hat\mu_m^k\sim\mathcal{N}(\mu_m^k, \hat{\Sigma}_{\mu_m^k}^2)$,
>
> and $\hat\sigma_m^k \sim \mathcal{N} (\sigma_m^k, \hat{\Sigma}^2_{\sigma_m^k})$, which are now consistent with Eq. 8.
>
> Beyond this typo, we notice that the symbol ${X}_m^{k}$ in Eq. 2 may cause confusions as well.
> We refine it to ${X}_m^{k,(h,w)}$ in the updated version to denote features at each spatial location $(h,w)$.
>
> ---
> We thank the reviewer again for confirming the originality of our work, as well as pointing out writing issues. We hope that the revisions/responses address your concerns, and look forward to your responses to this rebuttal.

---

### Official Review · Reviewer_JqCN · 2022-10-25

**Confidence:** 4
**Correctness:** 3
**Technical Novelty And Significance:** 3
**Empirical Novelty And Significance:** 2
**Recommendation:** 6

**Clarity, Quality, Novelty And Reproducibility:**

The writing is clear. The experiments design is generally good.
The proposed method is simple yet novel.

**Strength And Weaknesses:**

Strength:
1. The proposed method targets at the non-i.i.d. problem of FL, and the idea of feature level augmentation is interesting.
2. The FedFA method is simple but effective. Empirical comparisons with classical FL methods demonstrate the effectiveness of the proposed feature augmentation method.

Weakness:
1. FedRobust and FedBN also targets at feature shift problem in FL, it is not clear why "these algorithms may still suffer significant local dataset bias". Could the authors provide in-depth theoretical analysis how the proposed method is different/superior compared with these two counterparts? Besides, the experiments should show comparisons with FedRobust and FedBN.
2. I wonder if the method utilizes the statistics of features from all layers or specific layers. The paper lacks discussions and ablations regarding this.
3. The experiments only focus on splitting local data with different data origin. I would expect results with other kinds of local data heterogeneity, such as large variances among local data size or different modalities among locals.

**Summary Of The Paper:**

The paper proposes federated feature augmentation to solve the data heterogeneity in federated learning. Each local model models its feature with Gaussian distribution, where the variance is the key to fulfill augmentation.
The experiments on image classification and medical image segmentation show effectiveness of the proposed work.

**Summary Of The Review:**

The paper proposes a novel feature augmentation method for non-i.i.d. federated leaning.
My main concern is the method only shows effectiveness with local data split by data origin. There should be more experiments for other kinds of local data heterogeneity (e.g. variant local data size). A typical FL setting is to split data with Dirichlet distribution.

---

> ### Author Response · Authors · 2022-11-16
> **Response to Reviewer JqCN (1/2)**
>
> We thank the reviewer for the valuable feedback and suggestions. We begin the response by focusing on your main concern in following comments:
>
> > My main concern is the method only shows effectiveness with local data split by data origin.
> > There should be more experiments for other kinds of local data heterogeneity (e.g. variant local data size).
>
> > The experiments only focus on splitting local data with different data origin.
> > I would expect results with other kinds of local data heterogeneity,
> > such as large variances among local data size or different modalities among locals.
>
> As suggested, we provided extended results on two new datasets: CIFAR-10 and EMNIST.
> To fully eliminate your concern, we simulate two new kinds of data heterogeneity: **label distribution heterogeneity** in CIFAR-10
> and **data size heterogeneity** in EMNIST:
> * For CIFAR-10, we follow your advice to sample local data with heterogeneous label ratios using
>   Dirichlet distribution Dir$(\alpha)$, where $\alpha$ is set to $0.6$ or $0.3$ to simulate different non-i.i.d. levels, as [ref1, ref2].
> * For EMNIST, we follow FedMix to split data based on writers. This results in clients with varied numbers of local samples.
>
> The final experimental results show that FedFA performs favorably across different kinds of data heterogeneity.
> This confirms generality;  though designed for tackling feature shift non-i.i.d.,
> FedFA's very nature of data augmentation makes it a fundamental technique to handle various non-i.i.d. challenges in FL.
> Beyond empirical analysis, we offer theoretical analysis (as suggested) in following responses to
> explain the cause of generality from a federation-aware regularization perspective.
>
> |Algorithm | CIFAR-10, Dir(0.6) | CIFAR-10, Dir(0.3) | EMNIST|
> |:--------|:------------------:|:------------------:|:-----:|
> |FedAvg    |73.3 | 69.2 | 84.9|
> |FedAvgM   |73.4 | 69.1 | 85.5|
> |FedProx   |74.0 | 69.5 | 84.9|
> |FedBN     |73.7 | 69.8 | 85.3|
> |FedSAM    |74.3 | 70.0 | 86.5|
> |FedRobust |74.9 | 70.5 | 86.7|
> |FedMix    |75.5 | 70.7 | 86.6|
> |FedFA     |**76.3** |**71.9** |**87.8** |
>
>
> Regarding the counterparts FedRobust and FedBN, you mentioned:
> > FedRobust and FedBN also target at feature shift problem in FL,
> > it is not clear why "these algorithms may still suffer significant local dataset bias".
>
> As plainly stated in the 2nd paragraph of Introduction,
> these two methods are somewhat device-dependent:
> "FedRobust and FedBN solve the problem by either fitting the shift with a client-specific
> affine distribution or learning unique BN parameters for each client, respectively".
> As a result, they are more likely to overfit to biased local data distribution.
> In contrast, FedFA is  superior in that each client has access to universal statistic information
> of all participating clients, thus can potentially gain a better sense of the heterogeneity across clients.
> Through the globally-aware augmentation, each client is then trained to perform better on samples
> that may be different (in feature space) than their local data but that may be potentially present
> in other or even unseen clients.
> This helps to alleviate the effect of data heterogeneity and improve generality.
>
> In addition to intuitive insights, you asked for theoretical insights:
> > Could the authors provide in-depth theoretical analysis how the proposed method
> > is different/superior compared with these two counterparts?
>
> Sure. We have provided theoretical insights to FedFA in __Appendix B__.
> To summarize, we first interpret the approach as a federation-aware noise injection process.
> Then, we show that it imposes a natural form of regularization to local client training (Theorem 1)
> and derive the explicit form of the regularizer in the proof.
> Though regularization is known to intimately connect to noise injection [ref4, ref5],
> our Theorem 1 affords us novel insights that the regularizer in FedFA is federation-aware, in the sense that it is built on the universal statistic
> variances learned from the federation.
>
> In particular, for each client $m$, its local objective $\mathcal{L}_m^{\text{FedFA}}$ can be expressed as
> $\mathcal{L}_m^{\text{FedFA}} = \mathcal{L}_m^{\text{ERM}} + \mathcal{L}_m^{\text{REG}}$
> where $\mathcal{L}_m^{\text{ERM}}$ is the standard ERM loss in Eq. 1,
> and $\mathcal{L}_m^{\text{REG}}$ is the regularization term.
> We give the explicit form of $\mathcal{L}_m^{\text{REG}}$, and find that FedFA leads to
> regularization of the gradients of latent representations,
> with the regularization effects weighted by statistic variances,
> or more precisely, federation-aware noise defined in Eq. 11.
> _This regularization nature distinguishes our approach from FedRobust and FedBN,
> and makes our model theoretically more robust to local data heterogeneity._ Please refer to Appendix B for details.
>
> [Our response continues ...]

---

> > ### Author Response · Authors · 2022-11-16
> > **Response to Reviewer JqCN (2/2)**
> >
> > Moreover, you suggested:
> > > The experiments should show comparisons with FedRobust and FedBN.
> >
> > Agree. Their results are now included in Tables 1-4.
> >
> > We note that for FedBN, it is conceptually hard to generalize to new clients
> > without any fine-tuning; thus we omit its results in Table 4.
> >
> > > I wonder if the method utilizes the statistics of features from all layers or specific layers.
> > > The paper lacks discussions and ablations regarding this.
> >
> > By default, we add one federated feature augmentation layer (FFA) after each convolutional stage
> > of the networks. For instance, five FFA layers are added to AlexNet/U-Net,
> > as shown in Tables 11/15 in the appendix.
> >
> > To mitigate your concern, we study the sensitivity of FedFA to the set of eligible layers in Appendix C.
> > From the results, we observe that i) the default design always shows the best performance on the three datasets
> >   (Office, DomainNet and ProstateMRI).
> >   We conjecture that this is due to its potential to beget more comprehensive augmentation;
> > ii) applying FFA to only one particular stage brings minor gains against FedAvg;
> > but iii) by adding more layers, the performance tends to improve.
> > This implies that our approach benefits from inherent complementarity of
> > latent features in different network stages.

---

> > > ### Comment · Reviewer_JqCN · 2022-12-02
> > > **Communication cost and privacy concern**
> > >
> > > Is there any analysis of communication bandwidth or privacy guarantee compared with those purely parameter-sharing methods?
> > >
> > > Additional sharing of feature statistics not only has more communication cost, but also leads to more risks of private data leakage.  There should have a trade-off here. It will be more convincible if the table lists which kind of information each method shares (and the corresponding bandwidth, privacy risk), rather than only the acc.

---

> > > > ### Author Response · Authors · 2022-12-03
> > > > **Response to further communication cost and privacy concern (1/2)**
> > > >
> > > > Thanks for the further comments and suggestions.
> > > >
> > > > Regarding your question:
> > > >
> > > > > Is there any analysis of communication bandwidth or privacy guarantee compared with those purely parameter-sharing methods?
> > > >
> > > > The answer is a Yes for communication cost analysis,
> > > > as presented in Section 4.5 and reiterated here.
> > > > Concretely, we denote, for purely parameter-sharing methods (e.g., FedAvg),
> > > > the per-client per-round communication cost for transferal of model
> > > > parameters as $c_m$. Then, the corresponding cost of FedFA is $c_m + c_e$,
> > > > where $c_e=4\sum_{k=1}^KC_k$ is the extra cost induced by sharing feature statistics.
> > > > Here $K$ is the number of FFA layers used in the network, $C_k$ is the number of feature channels at layer $k$,
> > > > and the factor of $4$ is  for server receiving and sending two statistic values.
> > > > Note that in general we have $c_e\ll c_m$, e.g., for AlexNet $c_e=18$KB and $c_m=99$MB, for U-Net $c_e=15.5$KB and $c_m=59.2$MB.
> > > > _Hence, the extra communication overhead is in practice negligible. In the response to reviewer q8Hi,
> > > > we have also shown that the communication cost of FedFA is asymptotically same to FedAvg._
> > > >
> > > > For the aspect of privacy, we discuss more here and will integrate the discussions into the next revision.
> > > > First, we note that for modern neural networks with BN layers (which are common),
> > > > FedFA has a similar degree of privacy guarantees as
> > > > most purely parameter-sharing methods (excepting for FedBN that on purpose keeps BN locally without any aggregation).
> > > > Taking FedAvg as an example, in addition to aggregating model parameters,
> > > > the server receives local BN statistics (i.e., running mean and variance), computes their first moments (i.e., mean)
> > > > and then distributes them back to clients. FedFA is similar in that each client's momentumed local statistics (Eq.4) are sent to the server;
> > > > but slightly different in that we calculate the second moments (i.e., variance) as in Eq. 5.
> > > > Note that there is no direct local data sharing across clients in FedFA, making it privacy-aware superior to the counterpart
> > > > FedMix that exchanges averaged raw data across clients.
> > > >
> > > > Further, we point out that privacy is difficult to quantify and the notion itself is an active research topic in FL.  For example, even FedAvg suffers certain privacy issues, i.e., an adversary (e.g., one of the clients) can infer whether a sample belonging to other clients [ref8] or even precisely reconstruct their training data if gradients are shared [ref9]. A rigorous privacy analysis to FedFA is beyond the main scope of this work, but it will be our focus in the next stage. Thanks.
> > > >
> > > > [ref8] Nasr, Milad, et al. "Comprehensive privacy analysis of deep learning: Passive and active white-box inference attacks against centralized and federated learning."  IEEE symposium on security and privacy (SP) 2019.
> > > >
> > > > [ref9] Zhu, Ligeng, et al. "Deep leakage from gradients." NeurIPS 2019.
> > > >
> > > > > Additional sharing of feature statistics not only has more communication cost, but also leads to more risks of private data leakage. There should have a trade-off here.
> > > >
> > > > The trade-off can be very easily achieved in FedFA. Given a fixed neural network,
> > > > the amount of shared feature statistics in FedFA only depends upon the number of FFA layers used in the network.
> > > > Recall the ablative results in Table 9 (for addressing one of your previous concerns): using more layers generally leads to higher accuracy,
> > > > but at the cost of more information sharing (thus higher communication cost and possibly more risks of data leakage), and vice versa.
> > > > While we use by default a uniform of 5 FFA layers in all experiments for accuracy concern,
> > > > we can reduce the number to realize a better trade-off among accuracy,
> > > > communication cost or privacy risk, to meet specific requirements of various target applications.
> > > >
> > > > For example, as shown in Table 9, the model variant '{1,5}'
> > > > (with only two FFA layers)
> > > > can already lead to promising gains against FedAvg, i.e., +1.9%/2.7%/2.0% for Office/DomainNet/ProstateMRI.
> > > > Its extra communication cost is notably reduced to 5KB for AlexNet as compared to the 18KB under
> > > > the default configuration.

---

> > > > > ### Author Response · Authors · 2022-12-03
> > > > > **Response to further communication cost and privacy concern (2/2)**
> > > > >
> > > > > About the suggestion:
> > > > >
> > > > > > It will be more convincible if the table lists which kind of information each method shares (and the corresponding bandwidth, privacy risk), rather than only the acc.
> > > > >
> > > > > We agree. In the following table,
> > > > > we list for each algorithm the type of information it shares,
> > > > > along with its total communication cost for each client in a round.
> > > > > The table will be merged into Table 1 and Table 2 in the next revision to
> > > > > hopefully offer a more convinced and comprehensive comparision of the algorithms.
> > > > >
> > > > > |Algorithm |  Type of exchanging information | Cost per client (AlexNet) | Cost per client (U-Net) |
> > > > > |----------|:------------------:|:------------------:|:------:|
> > > > > |FedAvg | model parameters | 99 MB | 59.2 MB
> > > > > |FedProx |model parameters | 99 MB| 59.2 MB
> > > > > |FedSAM | model parameters | 99 MB| 59.2 MB
> > > > > |FedAvgM |model parameters | 99 MB| 59.2 MB
> > > > > |FedRobust |model parameters | 99 MB| 59.2 MB
> > > > > |FedBN | model parameters - BN parameters/statistics | 98.94 MB  | 59.16 MB
> > > > > |FedMix | model parameters + averaged raw data | 99 MB + 384 KB | 59.2 MB + 435KB
> > > > > |FedFA | model parameters + feature statistics | 99 MB + 18 KB | 59.2 MB + 15.5 KB
> > > > >
> > > > > _Side discussions._ The extra cost of FedMix depends on the spatial resolution of input data;
> > > > > thus it is unfriendly to dense prediction tasks (e.g., segmentation) that need high-resolution inputs to
> > > > > achieve good performance. FedFA is more favorable to diverse tasks since its extra cost only relies on
> > > > > feature channel numbers set in the networks.

---

### Official Review · Reviewer_uZsm · 2022-10-29

**Confidence:** 3
**Correctness:** 4
**Technical Novelty And Significance:** 4
**Empirical Novelty And Significance:** 3
**Recommendation:** 6

**Clarity, Quality, Novelty And Reproducibility:**

The paper is well motivated, and the proposed solution is conceptually simple and has good intuitions. The evaluations may be further improved by using the same datasets and model architecture in the original paper of FEDMIX.

The paper is clearly written, and it is very easy to read the paper.

The paper proposes a federated learning algorithm from a novel perspective of feature augmentation. Conceptually, it can bring additional advantages, compared to the SOTA work in this area (FEDMIX). However, the evaluations may be further improved by using the same datasets and model architecture in the original paper of FEDMIX.

**Strength And Weaknesses:**

Strength:
(1) The paper propose a federated learning algorithm from a novel perspective of feature augmentation.
(2) The proposed approach is conceptually simple but seems to be effective.


Weakness:
(1) The most related baseline seems to be FEDMIX. However, the evaluated datasets and the adopted model architecture in this paper are different from the datasets in the original paper of FEDMIX. There is limited discussion of the rationale of how the datasets are selected and why they are different from the original paper of FEDMIX.
(2) A key assumption of this paper is that the each feature statistic follows a multi-variate Gaussian distribution. I am not sure whether the assumption is valid, for the neural embeddings in general. The paper may discuss more details of how/when this assumption is satisfied in tasks relying on deep neural networks.
(3) Previous works (e.g., FEDMIX) provide theoretical discussions. This paper may be further improved by providing theoretical discussions.

**Summary Of The Paper:**

The paper develops a robust federated learning algorithm to address feature shift in clients’ samples, which can be caused by various factors, e.g., acquisition differences in medical imaging. The paper proposes FEDFA to tackle federated learning from a distinct perspective of federated feature augmentation. The paper models each feature statistic probabilistically via a Gaussian distribution, with the mean corresponding to the original statistic and the variance quantifying the augmentation scope, from which novel feature statistics can be drawn to fulfill augmentation.

**Summary Of The Review:**

The paper proposes a federated learning algorithm from a novel perspective of feature augmentation. Conceptually, it can bring additional advantages, compared to the SOTA work in this area (FEDMIX). However, the evaluations may be further improved by using the same datasets and model architecture in the original paper of FEDMIX. A key assumption of this paper is that the each feature statistic follows a multi-variate Gaussian distribution. The paper may discuss more details of how/when this assumption is satisfied in tasks relying on deep neural networks.

---

> ### Author Response · Authors · 2022-11-16
> **Response to Reviewer uZsm (1/2)**
>
> Thank you for your review. We have responded to your questions below, and wherever feasible, revised the paper to reflect these answers.
>
> To begin with, we focus on the following comment:
>
> > The most related baseline seems to be FedMix.
> > There is limited discussion of the rationale of how the datasets are selected and
> > why they are different from the original paper of FedMix.
>
> Indeed, FedFA and FedMix are related: they share a similar spirit of data augmentation in their solutions.
> However, the scientific questions addressed in the two works are not alike.
> As discussed in Introduction and you have also included in [summary of the paper],
> our work focuses on the problem of "feature shift non-IID", which is never studied in FedMix.
> This explains why we use different datasets than FedMix in original experiments,
> that is, to align with more relevant studies (e.g., FedBN [ref6] and FedHarmo [ref7])
> that focus on the same problem as ours, and render convinced validation/fair comparisons.
>
> In particular, for a fair and easy comparison, we strictly follow the setups in FedBN and FedHarmo.
> For _datasets_, two (i.e., Office and DomainNet) are from FedBN, and the remaining one (i.e., ProstateMRI) is from
> FedHarmo. These datasets cover diverse realistic scenarios of feature shifting (see Appendix D),
> and thus ideally match the problem of interest. For _networks_,
> we adopted the respective architectures for each dataset in FedBN and FedHarmo.
> We have tried our best to improve Section 4.1 to clarify this.
>
> Next we proceed to the suggestion:
> > The evaluations may be further improved by using the same datasets and model
> > architecture in the original paper of FedMix.
>
> As suggested, we consolidate the evaluations by providing results on CIFAR-10 and EMNIST that are used in FedMix.
> For EMNIST, we strictly adhere to the protocol in FedMix, while for CIFAR-10,
> we follow a more popular practice to simulate label distribution heterogeneity based on
> Dirichlet distribution $\text{Dir}(\alpha)$,
> where $\alpha$ is set to 0.6 or 0.3 to simulate different non-i.i.d. levels.
> Below table shows the new results, which have been integrated into Table 3 in the latest revision.
>
> |Algorithm | CIFAR-10, Dir(0.6) | CIFAR-10, Dir(0.3) | EMNIST |
> |----------|:------------------:|:------------------:|:------:|
> |FedMix    |75.5     | 70.7    | 86.6    |
> |FedFA     |**76.3** |**71.9** |**87.8** |
>
> FedFA brings clear and consistent improvements against FedMix across all settings:
> In CIFAR-10, FedFA surpasses FedMix by  **0.8%**/**1.2%** in Dir(0.6)/Dir(0.3); as the non-i.i.d. level increasing
> from Dir(0.6) to Dir(0.3), FedFA yields a
> larger gap of performance gain (0.8% &rarr; 1.2%), showing a strong
> capability in handling severer non-i.i.d. cases;
> in EMNIST, we see  a similar gain of **+1.2%**.
>
> _Based on these results, we can more safely conclude that FedFA outperforms FedMix._
>
>
> > A key assumption of this paper is that the each feature statistic follows a multi-variate Gaussian distribution.
> > I am not sure whether the assumption is valid, for the neural embeddings in general.
> > The paper may discuss more details of how/when this assumption is satisfied in tasks relying on deep neural networks.
>
> Thanks for your comments, but there might be a slight misunderstanding here (mainly due to our original description).
> Our approach does not hold _any_ assumption about feature statistic following a multivariate Gaussian distribution;
> instead, we only assume that each feature statistic is _augmented_ based on a multivariate Gaussian distribution.
> As shown in Eq. 9, the augmentation is in nature equivalent to adding Gaussian noise to the original feature statistic,
> but not necessarily yields a Gaussian-like feature statistic (letting alone feature) space.
>
> We guess the confusion may come from the Introduction where we say "we model each feature statistic in a probabilistic
> manner via a multi-variate Gaussian distribution";
> now we revise it to "we model _the augmentation of_ each feature statistic in a probabilistic manner
> via a multi-variate Gaussian distribution". Relevant statements are revised as well. Thanks.
>
> [Our response continues ...]

---

> > ### Author Response · Authors · 2022-11-16
> > **Response to Reviewer uZsm (2/2)**
> >
> > > This paper may be further improved by providing theoretical discussions.
> >
> > Thanks for your suggestion. We have made a lot of efforts to provide theoretical insights to the proposed approach,
> > as presented in __Appendix B__ (pages 13-16).
> > To summarize, we first interpret the approach as a federation-aware noise injection process
> > and then prove that it exhibits a natural form of implicit regularization to local client learning (see Theorem 1 on page 15).
> > Though regularization is known to intimately connect to noise injection [ref4, ref5],
> > our Theorem 1 affords us novel insights that the regularizer in FedFA is federation-aware, in the sense that it is built on the universal statistic
> > variances learned from the federation.
> >
> > In particular, for each client $m$, its local objective $\mathcal{L}_m^{\text{FedFA}}$ can be expressed as
> > $\mathcal{L}_m^{\text{FedFA}} = \mathcal{L}_m^{\text{ERM}} + \mathcal{L}_m^{\text{REG}}$
> > where $\mathcal{L}_m^{\text{ERM}}$ is the standard ERM loss in Eq. 1,
> > and $\mathcal{L}_m^{\text{REG}}$ is the regularization term.
> > We give the explicit form of $\mathcal{L}_m^{\text{REG}}$, and find that FedFA leads to
> > regularization of the gradients of latent representations,
> > with the regularization effects weighted by statistic variances,
> > or more precisely, federation-aware noise defined in Eq. 11.
> >
> > We refer the reviewer to Appendix B for the detailed theoretical analysis. Thanks.
> >
> > ---
> > We hope that the responses have sufficiently addressed your questions/concerns,
> > and believed that the revision has considerably strengthen our manuscript, both empirically and theoretically.
> > We look forward to your response to this rebuttal.

---

### Author Response · Authors · 2022-11-16
**General Response**

We thank reviewers for their valuable feedback.

We are encouraged that the reviewers unanimously recognize the novelty of our work.
Specifically, they find our paper well-motivated (uZsm) and have good originality (7jRY), our idea interesting (7jRY, JqCN, q8Hi), our solution simple yet novel (uZsm, JqCN, q8Hi) and technically solid (7jRY).
Reviewer q8Hi remarks that "the results can be tested", and the approach
will "definitely contribute to the body of knowledge in the domain/literature."
Reviewers uZsm, JqCN, q8Hi comment positively on paper writing.

In response to reviewers' comments and suggestions, we have revised our paper to hopefully address all of them, and point out main changes below for convenience.
1. We added results on two new datasets: CIFAR-10 and EMNIST, to address the major and shared concern by reviewers uZsm, JqCN, q8Hi. Please see Table 3. We will explain the results in separate responses to address each reviewer's specific comments;
2. We provided theoretical analysis for the approach in Appendix B, as suggested by reviewers uZsm and JqCN;
3. We carefully improved the writing of Section 3 to hopefully make it much clearer, based on valuable feedback from reviewer 7jRY;
4. We provided additional ablation study in Appendix C to study FedFA's sensitivity to the set of eligible layers to apply FFA, as per the suggestion of reviewer JqCN;
5. We provided a more formal analysis of computation/memory/communication costs of the proposed approach in Section 4.5, as suggested by reviewer q8Hi.
6. All of the typos, minor omissions, and unclear explanations pointed out by reviewers have been fixed.


We marked major updates in __red__ to make it easier to find the changes. More details and information of other changes can be found in the respective author responses for each reviewer.

---
_References cited in the response:_

[ref1] Qu, Zhe, et al. "Generalized Federated Learning via Sharpness Aware Minimization", ICML 2022.

[ref2] Kim, Jinkyu, et al. "Multi-Level Branched Regularization for Federated Learning", ICML 2022.

[ref3] Yoon, Tehrim, et al. "FedMix: Approximation of Mixup under Mean Augmented Federated Learning". ICLR 2021.

[ref4] Bishop, Chris M. "Training with noise is equivalent to Tikhonov regularization", Neural Computation 1995.

[ref5] Camuto, Alexander, et al. "Explicit Regularisation in Gaussian Noise Injections", NeurIPS 2020.

[ref6] Li, Xiaoxiao, et al. "FedBN: Federated Learning on Non-IID Features via Local Batch Normalization", ICLR 2021.

[ref7] Jiang, Meirui, et al. "HarmoFL: Harmonizing Local and Global Drifts in Federated Learning on Heterogeneous Medical Images". AAAI 2022.

---

### Public Comment · ~Debora_Caldarola1 · 2023-02-03
**On related previous works**

Dear Authors,

Congratulations on the acceptance of your very nice paper at ICLR 2023!

We would like to draw your attention to a few related earlier works that you might consider mentioning in your camera-ready version.

1. In our related work section, you mention papers using Sharpness-Aware Minimization  (SAM) in FL for improving generalization. In our work [1] - concurrent to [2] - we employ SAM optimizers on the client-side for handling heterogeneous data in FL. We also show that averaging weights with SWA (Stochastic Weight Averaging) on the server-side helps mitigating the issue of statistical heterogeneity, matching the centralized performance on CIFAR10-100 even in the cases of highest data heterogeneity (e.g., when the Dirichlet’s parameter alpha is equal to 0 or 0.5. See Table 6 and Figure 10). In addition, we address the issue of how some strong data augmentation techniques, such as Cutout and Mixup, fail at generalizing in extreme heterogeneous federated scenarios, which could be of interest for your analysis.
2. In [3], the authors propose using Gaussian mixtures for classifier calibration, which is related to your method.
3. Also, we point that [4], Liu et al 2020a is not using SAM optimizer for FL, as instead stated in your introduction.

Thanks in advance for considering our suggestions and taking into account our contribution.

[1] Caldarola, Debora, Barbara Caputo, and Marco Ciccone. "Improving generalization in federated learning by seeking flat minima." Computer Vision–ECCV 2022: 17th European Conference, Proceedings, Part XXIII. Cham: Springer Nature Switzerland, 2022.

[2] Qu, Zhe, et al. "Generalized federated learning via sharpness aware minimization." International Conference on Machine Learning. PMLR, 2022.

[3] Luo, Mi, et al. "No fear of heterogeneity: Classifier calibration for federated learning with non-iid data." Advances in Neural Information Processing Systems 34 (2021): 5972-5984.

[4] Quande Liu, Qi Dou, and Pheng Ann Heng. Shape-aware meta-learning for generalizing prostate mri segmentation to unseen domains. In MICCAI, 2020a.

---

> ### Author Response · Authors · 2023-02-04
> **thanks**
>
> Dear Debora,
>
> Thanks for your interests to our article.
>
> 1. Thanks for your referring the papers to us. We would be happy to discuss your ECCV 2022 paper [1] in the camera ready version to render a comprehensive discussion of model generalization in FL.
>
> 2. For [3], our idea is totally different from it: [3] models features by Gaussian mixture, while FedFA performs feature statistic augmentation based on Gaussian distribution, without assuming any prior distribution for features. We would like to include [3] in the final version as well to clarify this point.
>
> 3. Thanks for pointing this out this issue. This is a writing error. For the citation of [4] in the introduction, my intention is actually to cite [2].
>  We will correct this.

---

### Decision · Program_Chairs · 2023-01-20

**Decision:**

Accept: poster

**Justification For Why Not Higher Score:**

It's a simple idea, and it's well-deserved acceptance by showing good experimental results, but it didn't show any more surprising results than that.

**Justification For Why Not Lower Score:**

simple yet effective methods

**Metareview: Summary, Strengths And Weaknesses:**

This paper proposes an augmentation technique that can create robust features to the shift of client samples in a non-iid FL environment. It is a simple method that models the statistics of local features with a Gaussian distribution and augments by considering the statistics of neighboring clients in a FL environment.

Three reviewers agreed that the submission deals with important issues in the real FL environment and the proposed method is simple yet effective, and suggested accepting this paper. One reviewer originally suggested rejecting, but the authors sufficiently addressed the concerns raised by the reviewer through the rebuttal period, and this reviewer also is convinced.

I hope that the final version will solve all the issues raised by the reviewers, such as writing issues. In addition, the results of comparison with fedmix in datasets such as CIFAR, which were performed during the discussion period, should be further supplemented and included in the final version.

**Note From Pc:**

if the above contains the word "oral" or "spotlight" please see: "oral" presentation means -> notable-top-5% and "spotlight" means -> notable-top-25%. As stated in our emails, we are disassociating presentation type from AC recommendations